# Kv3.3 subunits control presynaptic action potential waveform and neurotransmitter release at a central excitatory synapse

Amy Richardson[1†], Victoria Ciampani[1], Mihai Stancu[2], Kseniia Bondarenko[1‡], Sherylanne Newton[1§], Joern R Steinert[1#], Nadia Pilati[3], Bruce P Graham[4], Conny Kopp-Scheinpflug[2], Ian D Forsythe[1*]

[1]Auditory Neurophysiology Laboratory, Department of Neuroscience, Psychology and Behaviour, College of Life Sciences, University of Leicester, Leicester, United Kingdom; [2]Division of Neurobiology, Faculty of Biology, Ludwig-Maximilians-University, Munich, Germany; [3]Autifony S.r.l., Istituto di Ricerca Pediatrica Citta'della Speranza, Corso Stati Uniti, Padova, Italy; [4]Computing Science and Mathematics, Faculty of Natural Sciences, University of Stirling, Stirling, United Kingdom

*For correspondence:
idf@le.ac.uk

Present address: [†]Department of Clinical and Experimental Epilepsy, Institute of Neurology, University College London, London, United Kingdom; [‡]Institute of Immunology and Infection Research, University of Edinburgh, Edinburgh, United Kingdom; [§]UCL Ear Institute, University College London, London, United Kingdom; [#]School of Life Sciences, Medical School, University of Nottingham, Nottingham, United Kingdom

**Summary:** Kv3 potassium currents mediate rapid repolarisation of action potentials (APs), supporting fast spikes and high repetition rates. Of the four Kv3 gene family members, Kv3.1 and Kv3.3 are highly expressed in the auditory brainstem and we exploited this to test for subunit-specific roles at the calyx of Held presynaptic terminal in the mouse. Deletion of Kv3.3 (but not Kv3.1) reduced presynaptic Kv3 channel immunolabelling, increased presynaptic AP duration and facilitated excitatory transmitter release; which in turn enhanced short-term depression during high-frequency transmission. The response to sound was delayed in the Kv3.3KO, with higher spontaneous and lower evoked firing, thereby reducing signal-to-noise ratio. Computational modelling showed that the enhanced EPSC and short-term depression in the Kv3.3KO reflected increased vesicle release probability and accelerated activity-dependent vesicle replenishment. We conclude that Kv3.3 mediates fast repolarisation for short precise APs, conserving transmission during sustained high-frequency activity at this glutamatergic excitatory synapse.

## Editor's evaluation

This work shows that Kv3.3 potassium channels play a major role in shaping the presynaptic action potential waveform of calyx-type auditory synapses. Mice that lack Kv3.3 showed auditory response deficits, including increases of spike latency and jitter. Overall, the study shows the uniquely important role for Kv3.3 channels in the fast synaptic transmission between the neurons that compute sound localization in mammals.

## Introduction

Kv3 voltage-gated potassium currents rapidly repolarise APs and underlie fast-spiking neuronal phenotypes, enabling high-frequency firing with temporal precision (*Rudy and McBain, 2001*; *Kaczmarek and Zhang, 2017*). Kv3 channels are expressed throughout the brain including the hippocampus, cortex, cerebellum, and auditory brainstem (*Weiser et al., 1994*; *Du et al., 2000*; *Lien and Jonas, 2003*). They influence dendritic integration (*Zagha et al., 2010*) and somatic AP waveform (*Espinosa et al., 2008*; *Rowan et al., 2014*; *Choudhury et al., 2020*), but Kv3 channels are also located in

particular axons, including nodes of Ranvier (*Devaux et al., 2003*) and synaptic terminals (*Schneggenburger and Forsythe, 2006*).

There are four Kv3 genes (*kcnc1-4*) specifying alpha subunits (Kv3.1–3.4) that assemble as tetramers (*Coetzee et al., 1999*; *Rudy et al., 1999*). The transmembrane domains are generally well conserved across subunits, but both the N- and C-terminal domains have diverse sequences, creating distinct inactivation kinetics, interaction with cytoskeletal proteins (*Blosa et al., 2015*; *Stevens et al., 2021*) and regulation by phosphorylation (*Macica et al., 2003*; *Song et al., 2005*; *Desai et al., 2008*). Kv3 channels display ultra-fast kinetics (*Grissmer et al., 1994*; *Labro et al., 2015*) with half-activation at positive voltages, making them particularly effective in AP repolarisation (*Brew and Forsythe, 1995*). N-type inactivation is a powerful means of regulating $K^+$ channel activity (*Hoshi et al., 1990*) and inactivation in the Kv3 family is influenced by the subunit(s) expressed and their regulation by protein phosphorylation (see *Kaczmarek and Zhang, 2017*).

Transgenic knockouts of one subunit generally show mild phenotypes, consistent with heterogeneous composition of native channels and functional redundancy (*Joho et al., 1999*; *Espinosa et al., 2001*; *Joho et al., 2006*). Co-expression of Kv3.1 and Kv3.3 has been widely observed in different brain regions (*Chang et al., 2007*) and we recently showed that principal neurons of the medial nucleus of the trapezoid body (MNTB) within the auditory brainstem, have Kv3 channels composed of Kv3.1 and Kv3.3 subunits. In the MNTB, in contrast to the lateral superior olive, each subunit type can compensate for deletion of the other (*Choudhury et al., 2020*).

Changes in AP waveform at the synaptic terminal critically control calcium influx and neurotransmitter release (*Borst and Sakmann, 1998*; *Forsythe et al., 1998*; *Yang et al., 2014*), this in turn influences short-term plasticity (*Sakaba and Neher, 2003*; *Hennig et al., 2008*; *Neher and Sakaba, 2008*; *Neher, 2017*; *Lipstein et al., 2021*) and presynaptic forms of long term potentiation at mossy fiber terminals (*Geiger and Jonas, 2000*). Indeed, tuning of voltage-gated sodium and potassium channel kinetics not only enhances fast signaling, but also increases metabolic efficiency (*Hu et al., 2018*). In the present study, we took advantage of the calyx of Held giant synapse in the MNTB to investigate the role of Kv3 subunits at that presynaptic terminal. These binaural auditory nuclei must rapidly integrate AP trains transmitted from left and right cochlea with microsecond accuracy (*Beiderbeck et al., 2018*; *Joris and Trussell, 2018*; *Karcz et al., 2011*). The region expresses Kv3.1 and Kv3.3 subunits, with little or no Kv3.2 and Kv3.4 (*Choudhury et al., 2020*). This synapse is optimised for speed and fidelity (*Taschenberger et al., 2002*) and is operating at the 'biophysical limit' of information processing, in that fast conducting axons and giant synapses with nano-domain localisation of P/Q $Ca^{2+}$ channels, combine with postsynaptic expression of fast AMPARs, short postsynaptic membrane time-constants and exceptionally rapid APs to enable binaural sound localisation (*Schneggenburger and Forsythe, 2006*; *Neher and Sakaba, 2008*; *Joris and Trussell, 2018*; *Young and Veeraraghavan, 2021*).

In this study, we test whether Kv3.1 or Kv3.3 subunits have a specific presynaptic role by examining the calyx of Held AP waveform and transmitter release on deletion of either subunit. We show that Kv3.3 subunits are localised to the presynaptic terminal and that their deletion increases transmitter release. We demonstrate the role of Kv3.3 in enhancing the speed and reliability of brainstem binaural auditory processing and conclude that presynaptic Kv3.3 subunits are essential for fast AP repolarisation at this excitatory synapse.

## Results

The experiments reported here were conducted using CBA/CrL wildtype mice or transgenic mice backcrossed onto CBA/CrL that lacked either Kv3.3 or Kv3.1 (the genotypes are referred to as WT, Kv3.3KO and Kv3.1KO). We previously reported that Kv3.1 and Kv3.3 mRNAs are highly expressed in the MNTB, and no compensation for either subunit in the respective knockouts was observed (*Choudhury et al., 2020*). The influence of these deletions was assessed in vivo using extracellular recording from the MNTB and in vitro using whole cell patch clamp from the calyx of Held and MNTB principal neurons. Additionally, immunohistochemical localisation of Kv3.1 and Kv3.3 was determined using expansion microscopy. Together, these methods were applied to determine the contribution of Kv3 subunits to the presynaptic AP waveform, to assess the impact on transmitter release and determine how this impacts the response to sound. As an additional control, we confirmed that both Kv3.1 and Kv3.3 mRNA are present in the cochlear nucleus where globular bushy cells give rise to the calyx of Held synaptic terminal. The percent contribution of each subunit gene to the cochlear

nucleus Kv3 mRNA was 30.3% ± 9.1% (Kv3.1), 7.7% ± 2.8% (Kv3.2), 52.0% ± 6.0% (Kv3.3), and 10.0% ± 1.6% (Kv3.4; for data see *Figure 1—figure supplement 1* and summary statistics). These values are similar to those measured in the MNTB (*Choudhury et al., 2020*). Additionally, immunohistochemistry confirmed that both spiral ganglion neurons and cochlear nucleus have broad neuronal staining for Kv3.1 and Kv3.3 and as validated using tissue from knockout mice (*Figure 1—figure supplements 2 and 3*).

## Kv3 channels contribute to action potential repolarisation at the calyx of Held terminal

Previous reports have employed potassium channel blockers (4-aminopyridine or tetraethylammonium) to show that Kv3 currents contribute to fast repolarisation of APs in the MNTB (*Forsythe, 1994*; *Brew and Forsythe, 1995*; *Wang et al., 1998*). Low concentrations of extracellular TEA (1 mM) give a relatively selective block of Kv3 potassium channels (*Johnston et al., 2010*). Whole terminal voltage-clamp recordings of the calyx terminal revealed an outward current, as shown in the current-voltage relations in *Figure 1A*, with 1 mM TEA blocking 28.7% of the total outward current at a command voltage of +10 mV (current amplitude in WT = 9.37 ± 2.6 nA, n=7 calyces; WT+TEA = 6.68±0.9 nA, n=6; unpaired t-test, p=0.037; mean ± SD). Injection of depolarising current steps (100 pA, 50ms) under current-clamp triggered a single AP in the presynaptic terminal, (*Figure 1B*) which increased in duration on perfusion of 1 mM TEA (*Figure 1C*; with inset showing ±TEA overlay). In tissue from WT mice the mean AP half-width increased from 0.28±0.02ms (control; n=9) to 0.51±0.1ms (n=7) in the presence of 1 mM TEA (*Figure 1E*), supporting the hypothesis that presynaptic Kv3 channels are present and contribute to AP repolarisation at the calyx of Held.

AP duration increased in terminals from Kv3.3KO mice, similar to that from WT mice in the presence of TEA. In contrast, the AP duration recorded from terminals of Kv3.1KOs was comparable to WT APs (see *Figure 1D and E*). AP half-widths increased from 0.28±0.02ms in WT (n=9) and 0.32±0.02ms Kv3.1KOs (n=5) to 0.43±0.03ms in Kv3.3KO mice (n=6) and 0.51±0.1ms in WT+TEA (n=7). This increase in duration at 50% amplitude was accompanied by a slowed rate of AP decay in both Kv3.3KO terminals and WT terminals upon perfusion of TEA (*Figure 1G*, one-way ANOVA, Tukey's post hoc, Kv3.3KO vs WT p=0.0042; TEA vs WT p=0.0016) with no changes in the rising phase of the AP (one-way ANOVA, p=0.50; *Figure 1H*), nor in the resting (input) membrane conductance (one-way ANOVA, p=0.56; *Figure 1I*) or AP threshold (one-way ANOVA, p=0.96; see statistics table). Similar significant changes in AP duration were observed at 25% and 75% amplitudes for deletion of Kv3.3, but deletion of Kv3.1 did not significantly change AP duration at 25%, 50%, or 75% amplitude (see *Figure 1—figure supplement 4*). This suggests that Kv3.3 subunits are of most importance for presynaptic Kv3 channels and fast AP repolarisation . Despite the lack of a significant increase in action potential duration measured in Kv3.1KOs, the rate of decay was significantly slowed from 99±20 mV/ms (n=10) to 66±16 mV/ms (n=5; one-way ANOVA, Tukey's post hoc, p=0.02), consistent with Kv3.1 having a secondary role in presynaptic AP repolarisation, as might be expected from their localisation at axonal nodes of Ranvier (*Devaux et al., 2003*) and potential to form heteromers with Kv3.3 subunits.

## Immunohistochemical localisation shows that presynaptic Kv3 channels require the presence of a Kv3.3 subunit

Since Kv3 channels are present in both the presynaptic calyx of Held and the postsynaptic MNTB neuron, conventional fluorescence immunohistochemistry fails to resolve any differential localisation of particular subunits in the presynaptic or postsynaptic membranes. An additional complication is that few studies have localised both Kv3.1 and Kv3.3 subunits, most focus on imaging one subunit (usually Kv3.1). Electron microscopic studies demonstrated that Kv3.1b is present in the membrane of the non-release face of the calyx of Held (*Elezgarai et al., 2003*), while *Wu et al., 2021* show that Kv3.3 is present at the release face of the calyx. Kv3.3 has also been demonstrated at terminals in the medial vestibular nucleus (*Brooke et al., 2010*) at the neuromuscular junction (*Brooke et al., 2004*) and in hippocampal mossy fibers (*Chang et al., 2007*). Hence, there is histological evidence for Kv3.1 and Kv3.3 at specific presynaptic terminals, but they are not universally expressed at all synaptic terminals.

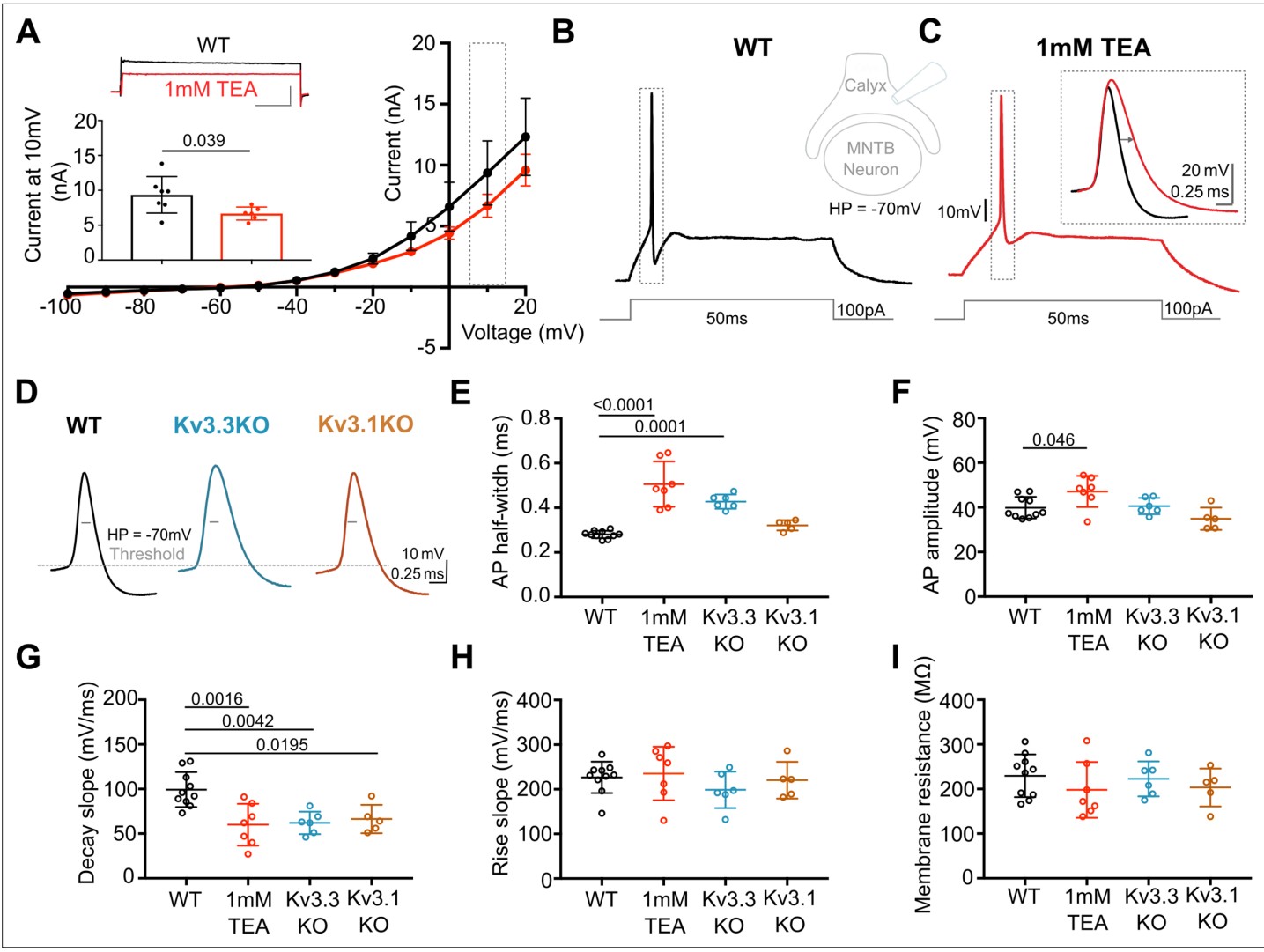

**Figure 1.** Presynaptic AP duration is increased by TEA or Kv3.3 deletion. (**A**) Current-Voltage (I–V) relationship for potassium currents in the calyx of Held terminal of WT mice (P10–P12) in control (black; n=7 terminals, 4 mice) and TEA, 1 mM (red, n=6 terminals, 5 mice; HP = –70 mV). **Inset (top):** Example current traces in response to voltage command of +10 mV step (grey box in IV) in WT (black) and WT +1 mM TEA (red). Scale bars = 5 nA and 20ms. **Inset (lower):** Bar graph of mean currents ± SD, measured on step depolarisation to +10 mV (from HP –70 mV) ±1 mM TEA. Outward Currents are significantly reduced by TEA (student's t-test, unpaired, P=0.0386). (**B**) WT calyx AP (black trace) evoked by 100 pA step current injection; **inset** - diagram of recording configuration. (**C**) WT calyx AP in the presence of TEA (1 mM, red trace); inset – overlaid WT APs ±TEA (red) as indicated by dotted box (grey) around APs in B and C. (**D**) Representative AP traces from calyx terminals of WT (black), Kv3.3KO (blue), and Kv3.1KO (orange); double arrows indicate the half-width of WT AP. AP threshold is indicated by the grey dashed line. (**E**) AP half-width measured as time difference between rise and decay phases at 50% maximal amplitude. Half-width is significantly increased in TEA and in Kv3.3KO; N is individual terminals: WT N=9 from 6 animals; TEA = 7 from 5 mice; Kv3.3KO = 6 from 3 mice and Kv3.1KO = 5 from 3 mice. (**F**) AP amplitude, (**G**) AP Decay slope, (**H**) AP rise slope (10–90%) and (**I**) membrane resistance for calyceal recordings. Average data presented as mean ± SD. Statistical test (parts E-I) were one-way ANOVAs and Tukey's post hoc for multiple comparisons, with significant Ps indicated on the graph.

The online version of this article includes the following source data and figure supplement(s) for figure 1:

**Source data 1.** Relates to *Figure 1*.

**Figure supplement 1.** WT mRNA Levels for *Kcnc* family genes in the cochlear nucleus which show a similar pattern of expression to those measured previously for the MNTB (*Choudhury et al., 2020*).

**Figure supplement 1—source data 1.** Relates to *Figure 1—figure supplement 1*.

**Figure supplement 2.** Kv3.1 and Kv3.3 immunoreactivity in the cochlear nucleus of CBA WT mice.

**Figure supplement 3.** Kv3.1 and Kv3.3 are present in spiral ganglion neurons of CBA WT mice.

*Figure 1 continued on next page*

Figure 1 continued

**Figure supplement 4.** Repolarisation of the presynaptic AP is slowed during the entire length of the downstroke in the presence of 1 mM TEA and in Kv3.3KOs.

**Figure supplement 4—source data 1.** Relates to *Figure 1—figure supplement 4*.

Capitalising on using KO-validated antibodies (*Choudhury et al., 2020 Figure 1—figure supplement 3*), we have employed protein-retention expansion microscopy (*Asano et al., 2018*) to increase resolution and gain more precision in localising Kv3.1 and Kv3.3 subunits at the calyx of Held synaptic terminal. The technique employs a hydrogel to expand the tissue by ~4.5 x prior to confocal imaging. We compared the location of Kv3 subunits in the three genotypes: antibodies targeting Kv3.3 were employed in the Kv3.1KO, antibodies against Kv3.1 were used in tissues from the Kv3.3KO and both Kv3.1 and Kv3.3 antibodies were employed in WT tissue. This experiment asks whether either Kv3.1 or Kv3.3 subunits are necessary and sufficient to enable presynaptic localisation of Kv3 channels. Synaptic release sites were identified using bassoon antibodies (*Chen et al., 2013*) to label the presynaptic specialisation and the results are shown in *Figure 2*.

In *Figure 2*, the two quadrants on the left show Kv3.3 staining (in the Kv3.1KO above, and WT, below) and the two right quadrants show Kv3.1 staining (in the Kv3.3KO above, and WT, below) in yellow. All sections are co-stained with bassoon (purple). Each quadrant shows single optical sections from three different MNTB neuron/calyx pairs shown at lower magnification in the respective top row. Kv3.3 staining was clearly present in the presynaptic terminal profiles surrounding each of the 3 MNTB neurons which lacked Kv3.1 (*Figure 2A1–C1*). In contrast, Kv3.1 labelling (from the Kv3.3KO tissue) showed little evidence of presynaptic labelling (*Figure 2D1–F1*). As a positive control, Kv3.1 labelling was robust in the postsynaptic membrane; and indeed Kv3.3 is also clearly present in the postsynaptic membrane. Each neuron in rows 1 and 4 has two synaptic profiles indicated by the orange and green arrowheads, each of these synapses are shown enlarged below their neuron of origin in rows 2 and 3 and rows 5 and 6, respectively. Similar labelling profiles are observed in the WT tissue (lower quadrants of *Figure 2A4–F4*). As for the KO tissue, two examples of WT presynaptic profiles are indicated by the coloured arrows and their enlargement shown below each principal neuron. The Kv3.3 antibody staining was clearly observed in the presynaptic membrane on the non-release face of the WT synapse (for example see *Figure 2B2 and C3, A6*) and in some image sections the release face of the synapse, between bassoon labelling (see *Figure 2C3, B5 and A6*) was clear. It was hard to visualise presynaptic membrane staining with the Kv3.1 antibody in the Kv3.3KO tissue (*Figure 2D2, E3 and F2*); but some Kv3.1-stained membrane profiles were observed on the non-release face in the WT tissue (*Figure 2E5, E6 and F6*).

While the calyceal Kv3.3 staining in the WT is similar to that in the Kv3.1KO, the levels of Kv3.1 staining in the WT are higher (than in the Kv3.3KO) consistent with the hypothesis that Kv3.1 may gain access to the presynaptic compartment through heteromeric assembly with Kv3.3 subunits. This data shows that while both Kv3.1 and Kv3.3 subunits are present in the postsynaptic soma membrane, only Kv3.3 subunits are required to achieve trafficking of Kv3 channels to the presynaptic terminal; and there is some evidence that the presence of Kv3.3 permits access of Kv3.1 subunits to the terminal, while both Kv3.1 and Kv3.3 were localised to the postsynaptic plasma membrane in the WT tissue.

## Deletion of Kv3.3 subunits increases calyceal-evoked EPSC amplitude

Transmitter release crucially depends on depolarisation of the presynaptic membrane and consequent calcium influx. Since Kv3 channels are present and involved in calyceal AP repolarisation, we assessed the physiological impact of each Kv3 subunit on transmitter release, and compared this to WT and to transmitter release following pharmacological block of presynaptic Kv3 channels using 1 mM TEA.

Whole cell patch recordings under voltage-clamp (HP = –40 mV, to inactivate voltage-gated Na$^+$ channels) were made from MNTB neurons possessing an intact calyx, in tissue from mice of each genotype and in the presence of 1 mM TEA. This allows comparison of the EPSC amplitude in four conditions: where Kv3 channels are intact (WT) and where they lack Kv3.1 or Kv3.3, and finally with all presynaptic Kv3 channels blocked (TEA). Unitary calyceal EPSCs were evoked in MNTB principal neurons as shown in *Figure 3A and F*. The global average for each condition and the unitary mean evoked EPSCs are overlaid from each genotype in *Figure 3F*. The dashed grey line shows the WT mean amplitude for comparison. The calyceal EPSCs recorded from Kv3.3KO mice were of larger peak

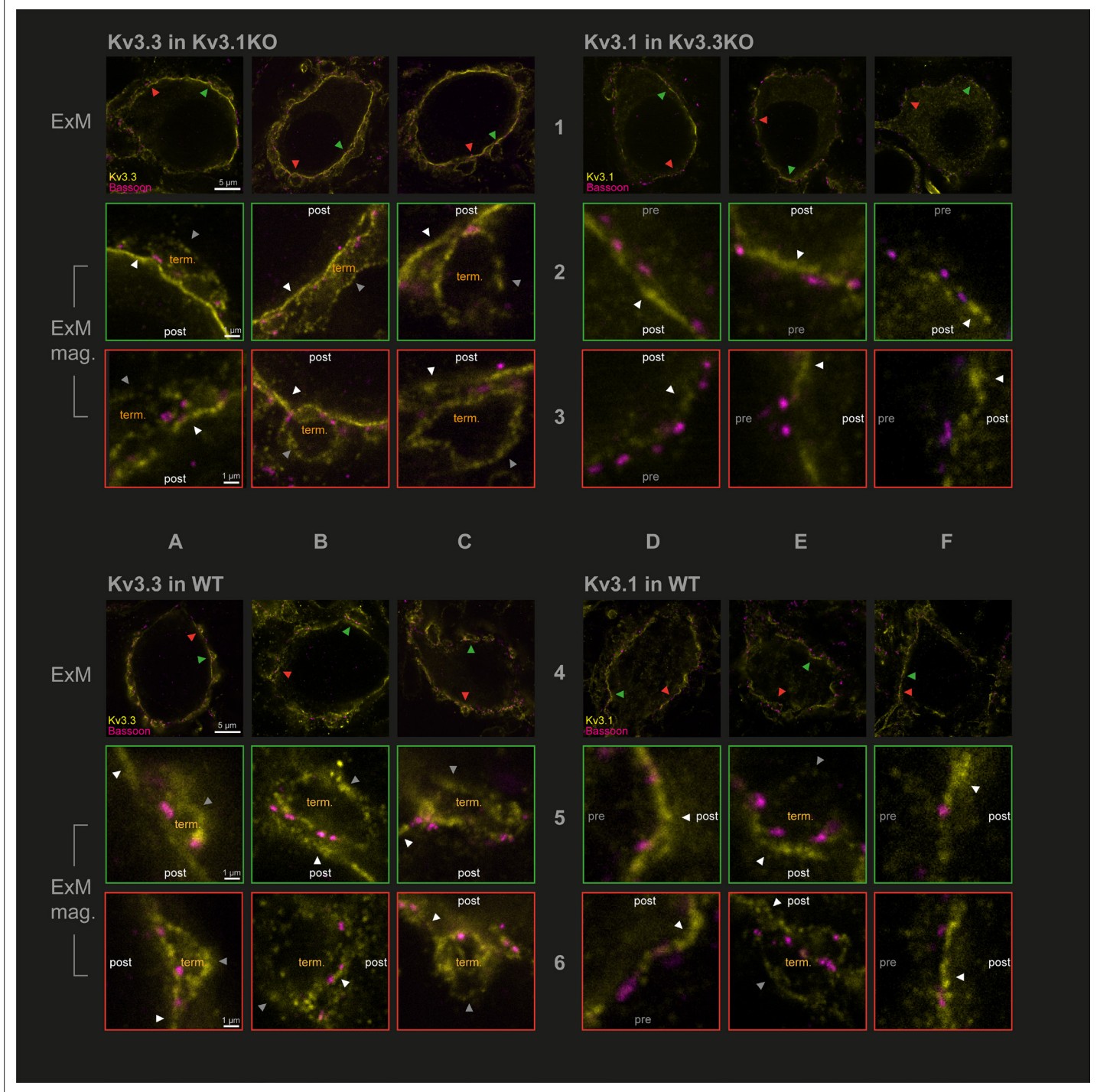

**Figure 2.** Protein-retention Expansion Microscopy (proExM) with confocal fluorescence imaging shows that Kv3.3 subunits are present in calyx of Held presynaptic terminal membrane. Individual images are identified in rows 1–6 and columns A-F, as indicated by the central labels. Four quadrants of 9 images are shown, each 3 × 3 matrix is from the named genotype and stained as specified in the title of each quadrant. The top row of each quadrant (rows 1 and 4) are single optical sections from 3 different MNTB neurons, in which their calyceal synaptic profiles are labelled with bassoon (purple) and co-labelled with either Kv3.1 or Kv3.3 antibodies (yellow): from Kv3.1KO and stained for Kv3.3 (**A1-C1**); the Kv3.3KO stained for Kv3.1 (**D1-F1**); WT stained for Kv3.3 (**A4-C4**); WT stained for Kv3.1 (**D4-F4**). In each MNTB neuron (rows 1 & 4) two synaptic regions of interest (ROI) containing bassoon are indicated by the red and green arrowheads. These magnified ROIs are displayed below (in rows 2+3 or 5+6) bordered by the same colour, respectively. The neuronal compartments are labelled: 'post' – postsynaptic; 'pre' – presynaptic; 'term' – synaptic terminal. In each image, the dark grey arrows point to presynaptic Kv3 labelling, and the white arrows point to postsynaptic Kv3 labelling. Scale bars are indicated for each row in column A (5 µm in rows 1 and 4: 1 µm in rows 2,3,5, and 6). Tissue was used from mice aged P28-P30.

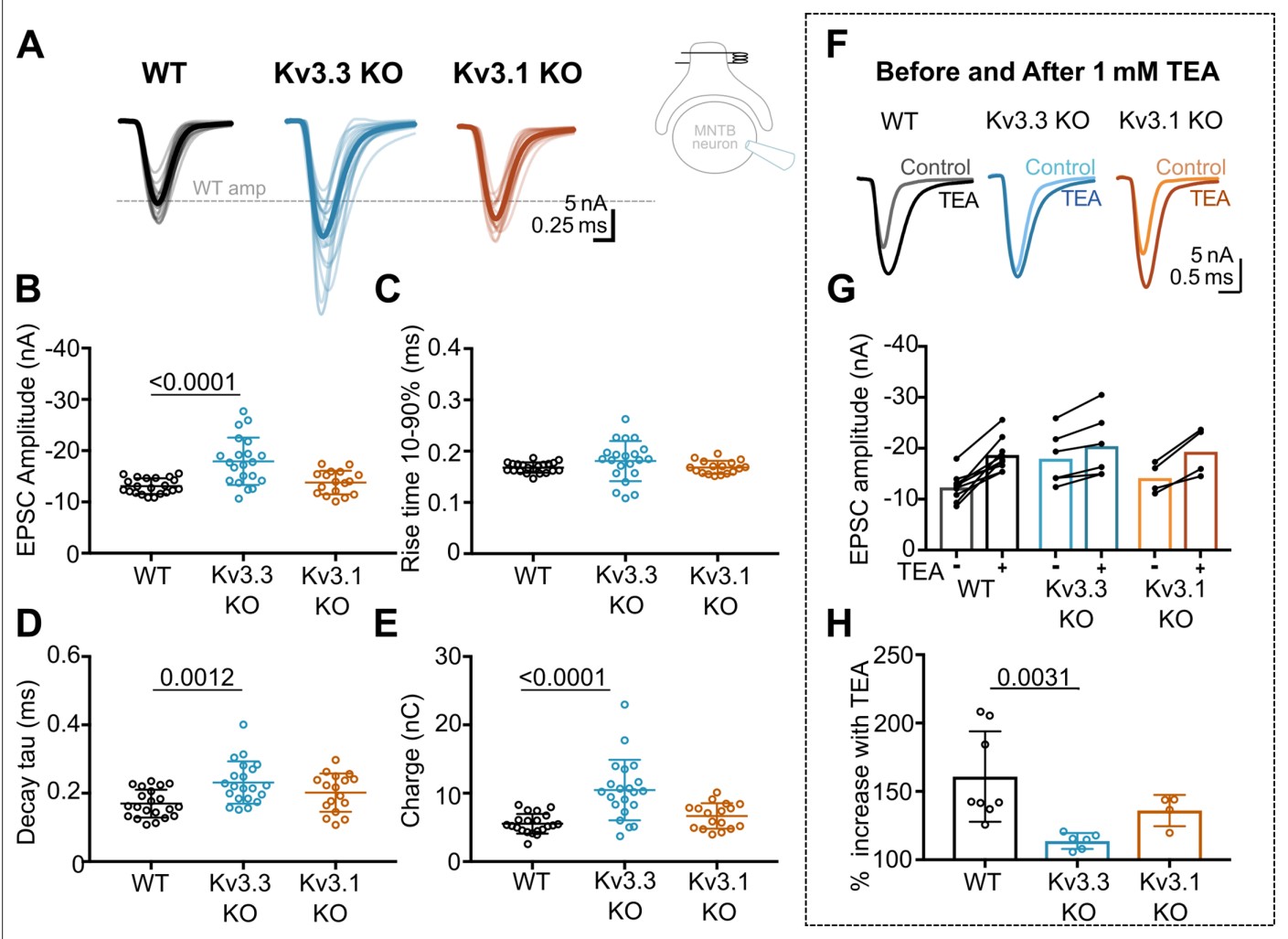

**Figure 3.** Kv3.3 deletion increases EPSC amplitude and occludes block by TEA. (**A**) Superimposed calyceal EPSCs generated from each genotype (age P21-P25): wildtype (WT; black), Kv3.3KO (blue), and Kv3.1KO mice (orange). Thin lines show traces from individual neurons (each is mean of 5 EPSCs) with thick line showing the population mean for each genotype. Grey dashed line indicates the average WT amplitude; N=WT, 22 neurons (11 mice); Kv3.3KO, 22 neurons (10 mice); Kv3.1KO, 17 neurons (8 mice). Inset shows recording and stimulation configuration. (**B**) EPSC amplitude increased in the Kv3.3KO. (**C**) EPSC rise time (10–90%) no difference was found between groups (one-way ANOVA, p=0.1576). (**D**) EPSC decay tau and (**E**) EPSC total charge were increased in the Kv3.3KO relative to WT. (**F**) EPSC traces from WT, Kv3.3KO, and Kv3.1KO mice, before and after the addition of 1 mM TEA. (Centre): EPSC amplitudes recorded before and after perfusion of TEA (1 mM); n=WT, 9 neurons (7 mice); Kv3.3KO, 6 neurons (3 mice); Kv3.1KO, 5 neurons (3 mice). (**G**) Increase in EPSC amplitude by 1 mM TEA. (**H**). The amplitude increase induced by TEA was significantly reduced in Kv3.3KO mice compared to WT. Average data presented as mean ± SD; statistics was using one-way ANOVA with Tukey's post hoc for multiple comparisons. Kruskal-Wallis ANOVA with Dunn's multiple corrections was used to compare change to EPSC amplitude in TEA due to a non-gaussian distribution in WT.

The online version of this article includes the following source data and figure supplement(s) for figure 3:

**Source data 1.** Relates to *Figure 3*.

**Figure supplement 1.** mEPSCs recorded from Kv3.3KO mice have a decreased amplitude.

**Figure supplement 1—source data 1.** Relates to *Figure 3—figure supplement 1*.

amplitude and longer lasting compared to WT or Kv3.1KOs, consistent with increased transmitter release from the calyx of Held in the Kv3.3KO.

The data for EPSC peak amplitude, rise-time, decay tau and charge are plotted in *Figure 3B–E*, respectively, for each genotype. EPSC amplitudes were similar in WT and Kv3.1KO, –13±2 nA (n=21) and –14±2 nA (n=17), respectively; but increased significantly in the Kv3.3KO to –18±5 nA (n=21; one-way ANOVA, Tukey's post hoc, Kv3.3KO vs WT, p=0.0001; Kv3.3KO vs Kv3.1KO p=0.0005). The decay tau for the EPSCs in the Kv3.3KO slowed to 0.23±0.06ms (n=21) compared to 0.17±0.04ms

in WT (n=21; one-way ANOVA, Tukey's post hoc, p=0.0012). The total charge of EPSCs in Kv3.3KOs increased to –10±4 nC (n=21) compared to –6±1 nC and –7±2 nC in WT (n=21) and Kv3.1KOs (n=17), respectively (one-way ANOVA, Tukey's post hoc; Kv3.3KO vs WT, p=0.0001; Kv3.3KO vs Kv3.1KO p=0.0007). No change of EPSC rise time was observed in either KO (one-way ANOVA, p=0.1918). Examination of miniature EPSCs in postsynaptic MNTB neurons from each genotype (*Figure 3—figure supplement 1*) showed no difference in the mEPSC frequency, rise-time or decay tau. There was a small significant decrease in quantal amplitude in the Kv3.3KO relative to WT, but no change in the Kv3.1KO. This contrasts with the increased amplitude of the evoked EPSCs in the Kv3.3KO measured here, and perhaps reflects an opposing change in synaptic scaling due to the larger transmitter release induced in the Kv3.3KO (*Tatavarty et al., 2013*).

The increased EPSC amplitude observed in Kv3.3KOs suggests that Kv3.3 subunits are major contributors to repolarisation of the presynaptic terminal. If Kv3.3 subunits dominate presynaptic Kv3 channels, then blocking presynaptic Kv3 channels with TEA will have a lesser effect on EPSCs from the Kv3.3KO. To test this hypothesis, we compared the effect of 1 mM TEA on EPSC amplitude from each of the three genotypes (*Figure 3F–H*). Indeed, blocking Kv3 channels with TEA had a much smaller effect on EPSC amplitude in the Kv3.3 KO compared to WT and Kv3.1 KO. TEA increased EPSC amplitude to 160% ± 33% (n=8) in WT, compared to 114% ± 6% (n=6) in Kv3.3KO (Kruskal-Wallis, Dunn's multiple comparison, p=0.0031). The EPSC amplitude in the Kv3.1KO (136% ± 11%, n=4) was not significantly different from WT (Kruskal-Wallis, Dunn's multiple comparison p=0.99). This result is consistent with dominance of presynaptic repolarisation by channels containing Kv3.3.

## Enhanced short-term depression in Kv3.3KO

The calyx of Held/MNTB synapse is capable of sustained transmission at firing frequencies of around 300 Hz in vivo (*Kopp-Scheinpflug et al., 2008*) (also see Figure 8), encoding information about sound stimuli for binaural integration with high temporal precision. Indeed, the calyx of Held giant synapse can sustain short AP bursts with peak firing rates of up to 1000 Hz, in vitro (*Kim et al., 2013*) for a few milliseconds. Clearly, the increased transmitter release observed in the Kv3.3KO (*Figure 3*) has consequences for maintenance of EPSC amplitude during repetitive firing, in that vesicle recycling and priming must rapidly replace docked vesicles if transmitter release is to be maintained for the duration of the high frequency train. Repetitive stimulation shows that the evoked EPSC amplitude declines during a stimulus train until rates of vesicle priming are in equilibrium with the rate of transmitter release for a given stimulus frequency. To assess repetitive transmitter release in each of the genotypes we evoked EPSCs over a range of frequencies from 100 to 600 Hz (with each stimulus train lasting 800ms).

Whole-cell patch recordings from voltage-clamped MNTB neurons (HP = –40 mV) were conducted in which calyceal EPSC trains were evoked at 100 Hz, 200 Hz, or 600 Hz. Each train was repeated three times, with a resting interval of 30 s between repetitions of stimuli trains. The EPSC amplitudes during the trains were compared in MNTB neurons of each genotype: WT (black), Kv3.3KO (blue) and Kv3.1KO (orange) mice. The first EPSC amplitude in trains delivered to the calyx/MNTB from the Kv3.3KO mouse was of larger amplitude than observed in WT or in the Kv3.1KO and subsequent EPSCs showed a larger short-term depression. In *Figure 4A–C* a matrix of EPSC trains are plotted with the same stimulus frequency in each row for each genotype. The inset traces show the first 3 EPSCs and the last 3 EPSCs in each train, with the black arrow heads indicating the WT amplitude for comparison across genotypes.

Paired pulse ratio (PPR) was significantly decreased in the Kv3.3KO compared to both WT and Kv3.1KO at 100 Hz, 200 Hz, and 600 Hz, with mean PPR ($EPSC_2/EPSC_1$) plotted in *Figure 4D* (left column). This enhanced depression of EPSC amplitude was maintained for the duration of the train shown in comparison with the 80th EPSC in the train ($EPSC_{80}/EPSC_1$; *Figure 4D*, right column; *Table 1*).

The mean data, normalized to the amplitude of the first EPSC, is plotted for each genotype at the stated stimulus frequency: *Figure 4E–G* (100–600 Hz). There are two key parameters plotted below each depression curve: first, the short-term depression decay time-constant (Decay Tau, defined as the rate at which the EPSC amplitude equilibrates to the new steady-state amplitude at each stimulus frequency); and second, amplitude at steady-state (the amplitude of the new steady-state EPSC during the train, relative to $EPSC_1$). The Kv3.3KO consistently showed the fastest rate of short-term depression, compared to WT and Kv3.1KOs; this was highly significant at 100 Hz and 200 Hz (*Table 1*).

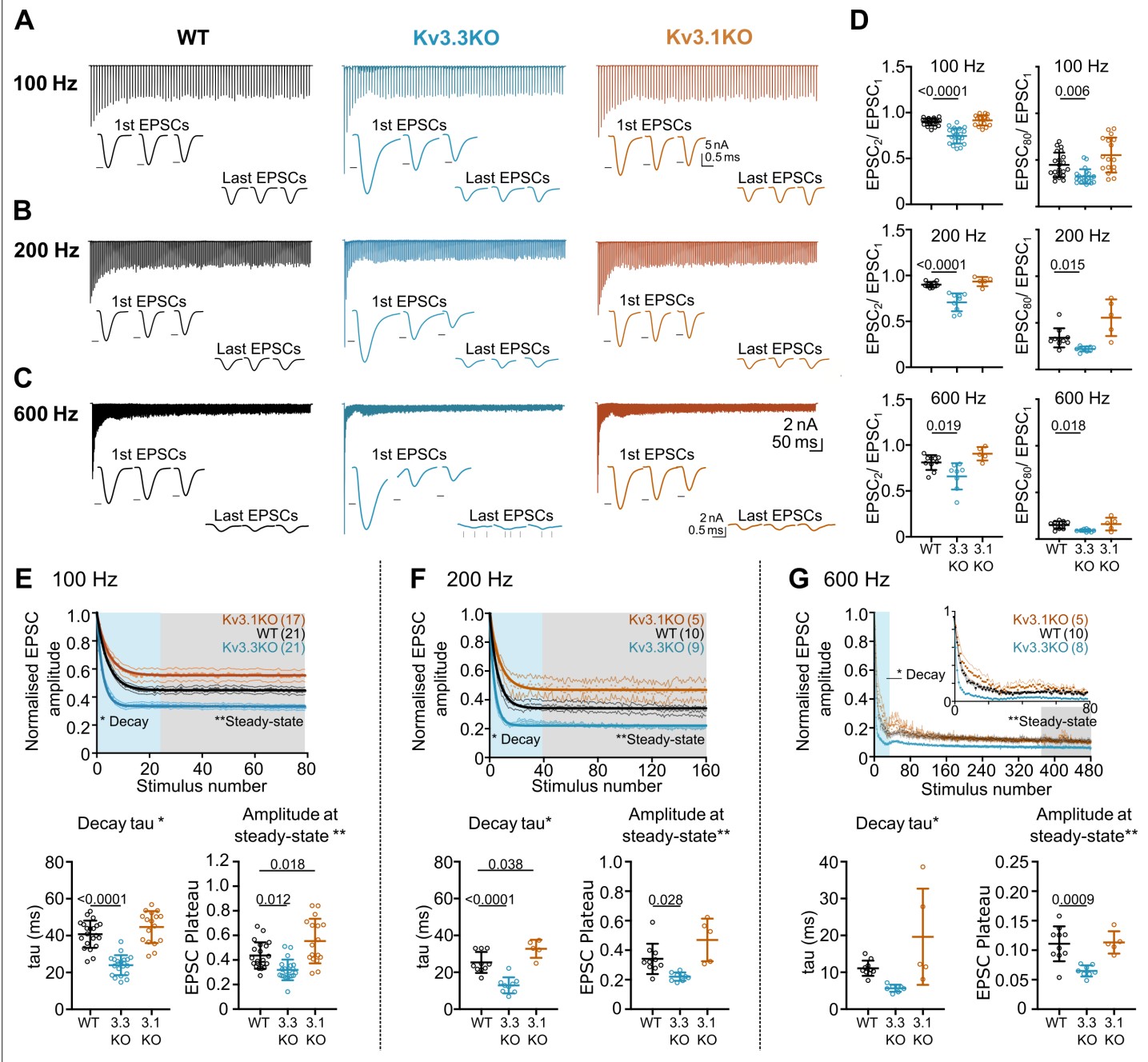

**Figure 4.** Evoked EPSC short-term depression is faster and more pronounced in the Kv3.3KO. (**A,B,C**) Example MNTB EPSC trains (800ms) on stimulation at (**A**) 100, (**B**) 200 or (**C**) 600 Hz for WT (left, black), Kv3.3KO (middle, blue), and Kv3.1KO (right, orange) mice (aged P21-P25). Each trace shows an average of 5 trials from a single neuron with stimulus artefacts removed for clarity. Lower insets: The first and last three EPSC traces are shown below each train. Black arrows show EPSC amplitudes from the WT mouse (left). Vertical grey arrows indicate multiple, asynchronous responses in the final EPSCs of Kv3.3KO traces. (**D**) Paired-pulse ratios for $EPSC_2/EPSC_1$ (left) and $EPSC_{80}/EPSC_1$ (right) generated in response to 100 Hz (top), 200 Hz (middle), and 600 Hz (bottom); in each case the ratio is significantly decreased in Kv3.3KO with respect to WT. For 100 Hz - WT: N=21 cells (11 mice); Kv3.3KO: N=21 (10 mice); Kv3.1KO: N=17 (8 mice); for 200 and 600 Hz - WT: N=10 (6 mice); Kv3.3KO: N=9 (4 mice); Kv3.1KO: N=5 (3 mice). Data shows mean ± SD and statistical significance estimated by one-way ANOVAs with Tukey's post hoc for multiple comparisons or Kruskal-Wallis with Dunn's multiple comparison (EPSC80/EPSC1 at 100 and 200 Hz due to non-gaussian data distributions), significant P values given on the respective graphs. (**E, F, G**) Normalised EPSCs, short-term depression is faster and larger in Kv3.3KO mice (compared to WT and Kv3.1KO mice) (**E**) 100 Hz F: 200 Hz G: 600 Hz. The rate of EPSC depression is plotted as a single exponential tau (lower left) and normalised EPSC amplitudes at steady-state are plotted (lower right) for each genotype and stimulus frequency. N numbers are presented in brackets on normalised EPSC amplitude graphs (and are the same neurons as

*Figure 4 continued on next page*

*Figure 4 continued*

used in D). One-way ANOVAs with Tukey's post hoc for multiple comparison correction were used to test significance, p values reported on graphs. All data plotted as mean ± SD, except in graphs of normalised EPSC amplitudes (E, F, and G, top) where data is plotted as mean ± SEM for clarity.

The online version of this article includes the following source data for figure 4:

**Source data 1.** Relates to *Figure 4*.

This trend continued so that during stimuli at 600 Hz (*Figure 4G*), a further increase in the rate of short-term depression was observed; again, this was most marked in EPSC trains from the Kv3.3KO compared to both WT and Kv3.1KO. The decay time-constant for short-term depression was 6±1ms for Kv3.3KOs (n=8), 11±2ms for WT (n=10) and 20±13ms for Kv3.1KOs (n=5; *Table 1*).

Following short-term depression, the 'steady-state' EPSC amplitude was achieved after 20–40 evoked responses (*Figure 4E, F and G*; lower right graph). This reflects the net equilibrium achieved under control of four key presynaptic parameters: probability of transmitter release, the rate of AP stimulation, the rate of vesicle replenishment at the release sites and the size of the vesicle pool (see modelling section below). In young animals, postsynaptic AMPAR desensitisation can also play a role in short-term depression, but this is a minor contribution at mature synapses and physiological temperatures, as employed here (*Taschenberger et al., 2002*; *Wong et al., 2003*). At all frequencies, EPSCs from the Kv3.3KO mice showed lower amplitude steady-state values in comparison to WT; while the Kv3.1KO data was either the same or greater than WT (*Figure 4E*, *Table 1*).

**Table 1.** Short-term depression was accelerated and enhanced in mice lacking Kv3.3.

Values shown are for parameters measured from data presented in *Figure 3* for WT, Kv3.3KO and Kv3.1KO genotypes at 100 Hz to 600 Hz range. Paired pulse depression of EPSC responses recorded in MNTB neurons (EPSC$_2$/EPSC$_1$) was increased in Kv3.3 KO animals during high frequency stimulation of the calyx. The increased depression was maintained throughout the stimulation train (EPSC$_{80}$/EPSC$_1$) across all frequencies. The rate of short term-depression in EPSC amplitudes during EPSC trains (duration 800ms), measured as short-term depression (STD) decay tau was significantly increased in Kv3.3 KOs at 100 and 200 Hz compared to WT. This STD was more severe in mice lacking Kv3.3, as shown by decreased normalized steady-state EPSC amplitudes compared to WT. STD tau and steady state amplitudes were measured using a single exponential fit to normalized EPSC amplitudes throughout the 800ms stimulation trains. n=number of neurons. Values in bold are significantly different to WT (see statistics table for more detail). Data represented as mean ± SD.

| | | WT | Kv3.3KO | Kv3.1KO |
|---|---|---|---|---|
| | EPSC$_2$/EPSC$_1$ Ratio | 0.90±0.04 (n=21) | **0.75±0.08 (n=21)** | 0.92±0.06 (n=17) |
| | EPSC$_{80}$/EPSC$_1$ Ratio | 0.45±0.13 (n=21) | **0.32±0.08 (n=21)** | 0.55±0.19 (n=17) |
| | STD tau (ms) | 41±7 (n=21) | **24±5 (n=21)** | 45±2 (n=17) |
| 100 Hz | Norm. steady-state amp | 0.4±0.1 (n=21) | **0.3±0.08 (n=21)** | **0.6±0.2 (n=17)** |
| | EPSC$_2$/EPSC$_1$ Ratio | 0.90±0.03 (n=10) | **0.71±0.1 (n=9)** | 0.93±0.05 (n=5) |
| | EPSC$_{80}$/EPSC$_1$ Ratio | 0.34±0.10 (n=10) | **0.22±0.03 (n=9)** | 0.56±0.20 (n=5) |
| | STD tau (ms) | 25±6 (n=10) | **13±4 (n=9)** | **33±5 (n=5)** |
| 200 Hz | Norm. steady-state amp | 0.3±0.10 (n=10) | **0.2±0.03 (n=9)** | 0.5±0.1 (n=5) |
| | EPSC$_2$/EPSC$_1$ Ratio | 0.91±0.06 (n=8) | **0.68±0.10 (n=9)** | 0.91±0.15 (n=6) |
| | EPSC$_{80}$/EPSC$_1$ Ratio | 0.25±0.19 (n=8) | **0.1±0.02 (n=9)** | 0.18±0.042 (n=6) |
| | STD tau (ms) | 7±5 (n=7) | 3±1 (n=6) | 5±2 (n=4) |
| 400 Hz | Norm. steady-state amp | 0.26±0.12 (n=6) | **0.12±0.04 (n=7)** | 0.12±0.04 (n=3) |
| | EPSC$_2$/EPSC$_1$ Ratio | 0.8±0.08 (n=10) | **0.7±0.1 (n=8)** | 0.9±0.07 (n=5) |
| | EPSC$_{80}$/EPSC$_1$ Ratio | 0.15±0.04 (n=10) | **0.09±0.01 (n=8)** | 0.16±0.07 (n=5) |
| | STD tau (ms) | 11±2 (n=10) | 6±1 (n=8) | 20±13 (n=5) |
| 600 Hz | Norm. steady-state amp | 0.11±0.03 (n=10) | **0.07±0.01 (n=8)** | 0.11±0.02 (n=5) |

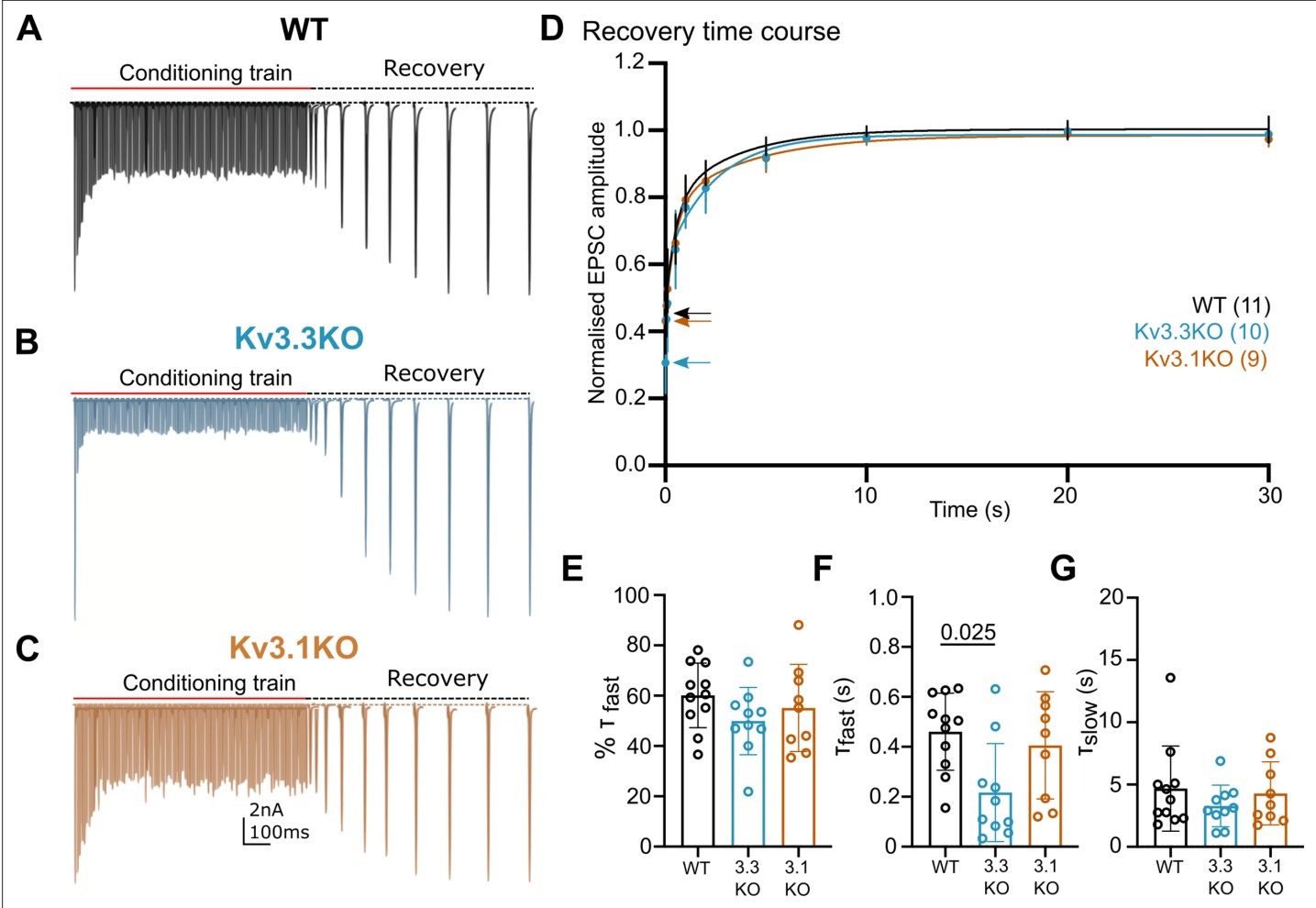

**Figure 5.** Recovery from short-term depression is accelerated on deletion of Kv3.3. (**A**) WT (black),(**B**) Kv3.3KO (blue),(**C**) Kv3.1KO (orange). A representative example is shown for each genotype. A conditioning train of 100 Hz (800ms duration) evoked EPSCs displaying short-term depression. The recovery was estimated by delivery of single stimuli at intervals following the conditioning train (50ms, 100ms, 500ms, 1 s, 2 s, 5 s, 10 s, 20 s, and 30 s. Recovery intervals not to scale).(**D**) The mean EPSC amplitude during the recovery is plotted for each genotype (mean ± SD. WT, black; Kv3.3KO, blue; Kv3.1KO, orange) over the 30 s recovery period. The mean amplitude at the end of the conditioning train, from which the recovery starts, is shown by the respective coloured arrow. A double exponential was fit to each individual recovery curve and the mean curve is plotted for the respective genotype. Values are plotted as raw data and mean ± SD in E-G; N values from part D also apply here. (**E**) The percent contribution of the fast component is similar between genotypes. (**F**) The fast time-constant significantly accelerated from 0.4 s in WT to 0.2 s in the Kv3.3KO. (**G**) The slow time-constant at around 5 s was unchanged between genotypes.

The online version of this article includes the following source data for figure 5:

**Source data 1.** ,Relates to *Figure 5*.

## Kv3.3 deletion accelerated a fast component of recovery from short-term depression

Presynaptic spike broadening has previously been shown to enhance vesicle recycling during repetitive stimulation trains (*Wang and Kaczmarek, 1998*) due to enhanced calcium-dependent recovery. An increase in the rate of recovery of the evoked EPSC following short-term depression (*Figure 5*) could indicate changes in vesicle recycling. In WT animals the EPSC was depressed to around 40% on stimulation at 100 Hz (*Figure 4E*), increasing to 90% depression for 600 Hz (*Figure 4G*). Three example conditioning traces are shown in *Figure 5A–C* for each genotype. On ceasing stimulation, the depressed EPSC recovered back to control amplitudes over a time-course of 30 s (*Figure 5A–C*, recovery). The recovery phase was measured by presenting stimuli at intervals of: 50, 100, 200, 500 ms, and 1, 2, 5, 10, 20, and 30 s after the end of the conditioning train. Each recovery curve was

repeated 3 times and the mean EPSC amplitudes plotted over the 30 s period (*Figure 5D*) and fit with a double exponential. The fast time-constant contributed around half of the recovery amplitude and this did not differ significantly between the three genotypes (*Figure 5E*). The fast time-constant was significantly accelerated in the Kv3.3KO compared to WT and Kv3.1KO (0.2±0.2 s in Kv3.3KO; 0.45±0.2 s in WT and 0.4±0.2 s in the Kv3.1KO; *Figure 5F*, p=0.025). The slow time-constant was 5±3 s in WT, and was similar to both Kv3.1KO and Kv3.3KO (*Figure 5G*). The enhanced fast recovery rate is consistent with an activity-dependent component of vesicle recycling as observed previously on blocking presynaptic Kv3 channels at the calyx of Held (*Wang and Kaczmarek, 1998*) and as recently attributed to $Ca^{2+}$-phospholipid-dependent vesicle priming (*Lipstein et al., 2021*) via Munc13-1.

## Computational model of transmitter release

The magnitudes and rates of EPSC depression and recovery following synaptic train stimulation provided constraints in refining a computational model of transmitter release and vesicle recycling at the calyx of Held (*Graham et al., 2004*). A simple model of transmission at the calyx of Held, based on transmitter release and recycling was employed as set out in *Figure 6A*, *equations 1-3*. It included activity-dependent vesicle recycling (*Graham et al., 2004*; *Billups et al., 2005*; *Lucas et al., 2018*) and parameters were fit across the range of stimulus frequencies (100–600 Hz) for the WT and the Kv3.3KO mouse. The model possessed a readily releasable pool (RRP; normalized size n), from which vesicles undergo evoked exocytosis with a release probability (Pv), following each AP and are recycled with a time-constant $\tau_r$. A basal rate of recycling (large time-constant $\tau_b$) is accelerated to an activity-dependent rate (smaller time-constant $\tau_h$) on invasion of an AP, and relaxed back to $\tau_b$ with a time-constant of $\tau_d$. The table in *Figure 6B* shows that the dominant change in the model parameters between WT and Kv3.3KO, was an increase in the probability of vesicular release (Pv) from 0.13 in WT to 0.266 in the Kv3.3KO. There was also evidence for an increase in activity-dependent recycling, with a 20% acceleration of the replenishment rate (smaller $\tau_h$ in Kv3.3KO). The fit of the model to the mean experimental data (± SEM) is shown for the short-term depression and the recovery curves (WT: *Figure 6C–D*; Kv3.3KO: *Figure 6E–F*). This model showed that the physiologically observed changes in short-term depression and recycling could be fit across the range of stimuli rates with only two parameter changes in the Kv3.3KO: a dramatically increased Pv and a modest decrease in the activity-dependent vesicle recycling time-constant $\tau_h$ (higher recycling rate), both of which are attributable to the raised calcium influx during the presynaptic AP (see *Neher and Sakaba, 2008*; *Young and Veeraraghavan, 2021*).

## Integration of EPSCs in generating APs in the postsynaptic MNTB neuron

A key physiological question is the extent to which presynaptic Kv3.3 influences AP firing of the MNTB neuron in response to trains of synaptic stimuli, since most auditory stimuli will be trains rather than single APs. This was addressed in two experiments: first in an in vitro slice study comparing the AP firing of MNTB neurons in response to synaptic stimulation over a range of frequencies and across the genotypes (*Figure 7*) and then through an in vivo study of the response of MNTB neurons to sound-evoked inputs, focusing on the Kv3.3KO and WT genotypes (*Figure 8*).

The first question was how does MNTB AP firing in response to a train of synaptic inputs change on deletion of Kv3.1 and Kv3.3? The initial observation was that MNTB AP output in response to high frequency calyceal EPSC input, declines with both Kv3.1 and Kv3.3 deletion (*Figure 7A*) when measured over the whole train (WT: 52.49% ± 8.15%, n=5; Kv3.3KO: 29.8% ± 10.43%, n=7; Kv3.1KO: 36.29% ± 11.12%, n=6; Kv3.3KO vs WT p=0.0044, Kv3.1KO vs WT p=0.046, one-way ANOVA, Tukey's post hoc). But closer inspection reveals three phases of AP firing to evoked trains of calyceal synaptic responses as illustrated in *Figure 7B*. This shows MNTB EPSPs/APs during an 800ms 600 Hz train in a WT mouse. In Phase I (green trace, *Figure 7B*) which predominates at the start of each train, every evoked EPSP triggered one MNTB AP: so the MNTB output matched the calyx input. Firing then transitioned to Phase II (blue trace, *Figure 7B*) after around 6–9 stimuli, where EPSPs often failed to evoke an AP, and the MNTB AP firing becomes chaotic, unpredictable and poorly transmits the timing information contained within the presynaptic AP train. In Phase III (black trace, *Figure 7B*): the EPSP had undergone short-term depression and approached a 'steady-state' amplitude; now the MNTB neuron fired APs to alternate EPSPs. At frequencies up to 200 Hz the duration of Phase I was essentially

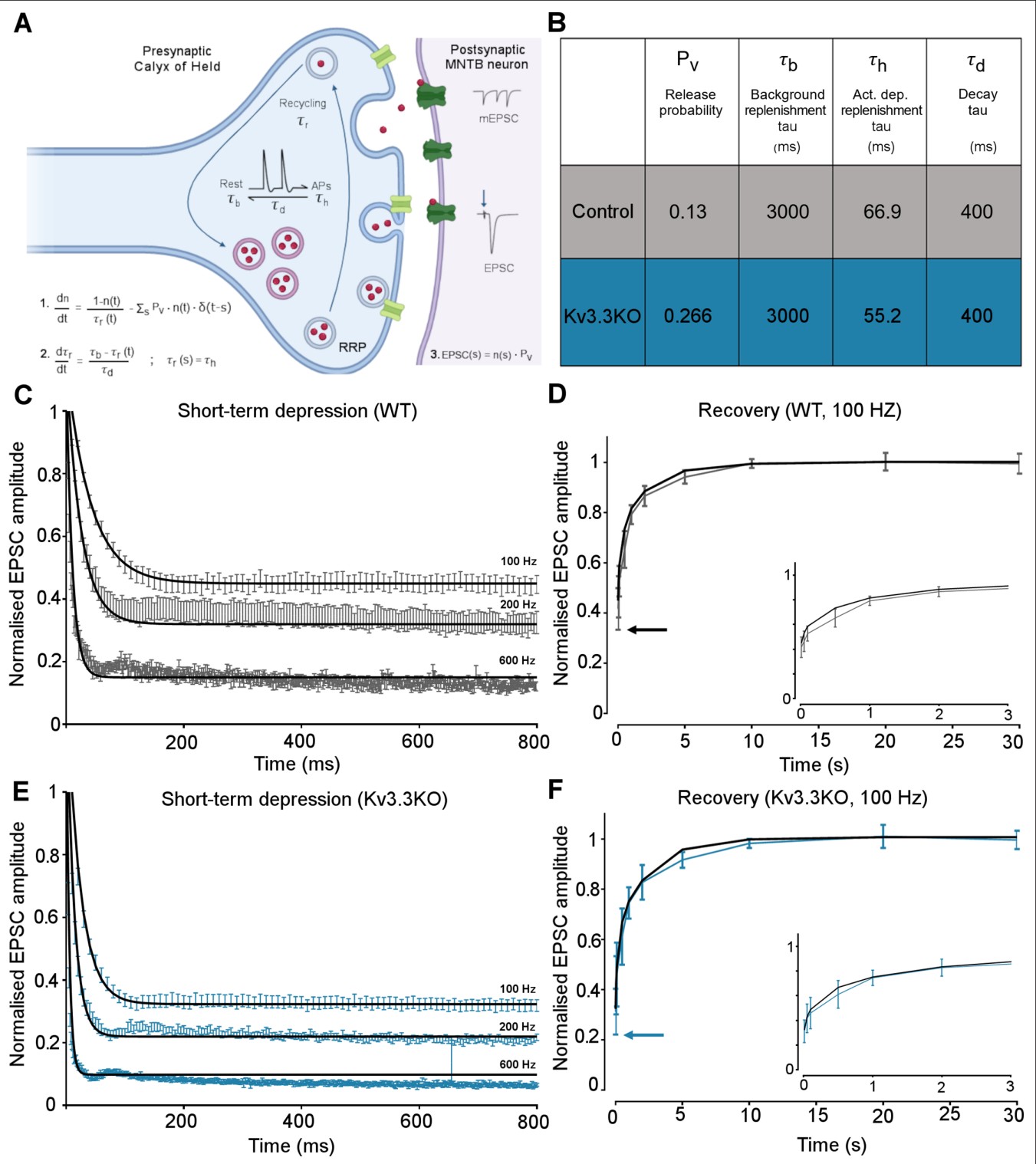

**Figure 6.** Kv3.3 deletion increases release probability and speeds a fast component of vesicle recycling, based on computational modelling. (**A**) Model illustration: Vesicles are released from the readily releasable pool (RRP; normalized size n) with a probability Pv; the RRP is replenished with the recycling time-constant $t_r$. In the absence of APs $\tau_r = \tau_b$ – the background replenishment time-constant, which decreases (accelerates) to $\tau_h$ following a presynaptic AP, and decays back to $\tau_b$ with a time-constant $\tau_d$. Model parameters were fit to WT data for evoked EPSC trains (100–600 Hz) and their 100 Hz recovery curves. Schematic created with [BioRender.com](BioRender.com). (**B**) Table showing the model parameters for fitting WT and Kv3.3KO data. Increasing Pv from 0.13 to 0.266 and accelerating $\tau_h$ from 66.9 to 52.2 were sufficient to fit the changes observed on Kv3.3 deletion. (**C**) WT - EPSC amplitude data (mean ± SEM) during the conditioning train (100, 200, and 600 Hz, grey) are plotted with superimposed model prediction curves (black). (**D**) WT

*Figure 6 continued on next page*

*Figure 6 continued*

100 Hz: Recovery of the EPSC (mean ± SEM) over 30 s. Inset shows data and fit for the first 3 s. Model fit is the superimposed black line. Horizontal arrow indicates EPSC amplitude at the end of the conditioning train. (**E** )Kv3.3KO: EPSC amplitude data (mean ± SEM) during the conditioning trains are plotted (blue) with superimposed model prediction curves (black). (**F**) Kv3.3KO 100 Hz: Recovery of the EPSC amplitude (mean ± SEM) over 30 s. Inset shows data and fit for the first 3 s. Model fit is the superimposed line. Horizontal arrow indicates EPSC amplitude at the end of the conditioning train.

The online version of this article includes the following source data for figure 6:

**Source data 1.** Relates to *Figure 6*.

identical across the three genotypes (*Figure 7C*), all showed 100% firing throughout the train; then from 300 Hz and above, the Phase I duration declined dramatically for all genotypes. Genotype-specific limitations were observed at the highest frequencies. *Figure 7D* shows the relative duration of each phase for 600 Hz EPSP trains (as a % of the train duration). There were no differences in the duration of Phase I, while Phase II and III were of variable duration. At 600 Hz stimulation frequency, the latency to the start of Phase II was 34.3±12.3ms for WT (n=10), 25±8ms for Kv3.3KO (n=6) and 26±14ms for Kv3.1KO (n=5). The latency for Phase III was similar in WT and Kv3.1KO (510±224ms and 597±251ms, respectively) but in the Kv3.3KO only 1 out of 6 calyx/MNTB pairs briefly entered Phase III firing (Kv3.3KO vs WT Phase II p=0.0135, Phase III p=0.02, two-way ANOVA, Tukey's post hoc). This is consistent with the idea that presynaptic Kv3.3 and hence fast presynaptic APs, serve a role in maintaining information transmission across the synapse during high frequency firing, when short APs conserve resources, and slow the rate of short-term depression.

Large magnitude Kv3 currents in postsynaptic MNTB neurons demonstrably assists in transmission of timing information (*Song et al., 2005*). Kv3 has little impact on the resting MNTB neuron membrane time-constant or AP firing threshold (induced by current injection through the pipette) and these are essentially identical in the three genotypes (*Choudhury et al., 2020*), which was also confirmed here (*Figure 7i*). However, in the Kv3.1KO, calyceal stimulation evoked a sustained depolarisation (in addition to EPSPs) at frequencies above 100 Hz (*Figure 7E*). The mean amplitude of this depolarising plateau potential during the train is plotted against stimulus frequency for each genotype (*Figure 7F*). Although all genotypes exhibited this plateau depolarisation at 600 Hz, it was significantly larger for the Kv3.1KO at frequencies above 100 Hz (*Figure 7F*). In contrast, Kv3.3KOs showed a reduced plateau potential at frequencies above 300 Hz, reaching significance only at 600 Hz (*Figure 7F*; two-way ANOVA, Tukey's post hoc). This was also observed as a decaying depolarisation at the end of the train (*Figure 7G*), with the time to half-decay plotted in *Figure 7H* (WT: 2.79±1.73ms, n=7: Kv3.3KO: 1.89±0.92ms, n=6: Kv3.1KO: 15.96±8.59ms, n=8: Kv3.1KO vs WT p=0.000553, one-way ANOVA, Tukey's post hoc). In WT and Kv3.3KO genotypes, this decay was similar to the postsynaptic membrane time-constant, but in the Kv3.1KO, non-synchronous spontaneous EPSPs were observed. The EPSC decay time constants for all genotypes were essentially identical, consistent with no change in glutamate receptor expression (WT: 2.74±1.26ms, n=8; Kv3.3KO: 3.1±0.87ms, n=8; Kv3.1KO: 3.54±0.9ms, n=6). This plateau depolarisation caused increased depolarisation block of APs and thereby undermined our ability to examine AP firing physiology in the Kv3.1KO. Therefore, in vivo experiments focused on comparison of the WT and Kv3.3KO genotypes, which did not exhibit this postsynaptic epi-phenomenon.

## Kv3.3 increases precision and signal-to-noise ratios in MNTB response to sound

The prolonged presynaptic AP duration and increased transmitter release observed here in the Kv3.3KO, shows that the calyx has a specific need for Kv3.3 channels. So, what is the impact of presynaptic Kv3.3 on brainstem auditory processing in the MNTB?

Behaviourally, the Kv3.3KO mouse is as sensitive to sound as the WT in that Auditory Brainstem Response (ABR) thresholds were similar in 6-month-old WT and Kv3.3KO mice, with no statistical difference across a wide frequency range (see summary statistics, unpaired t-test with Holm-Šídák's test for multiple comparisons). However, further analysis of the ABR waveform showed significant deficits in Wave IV at 6 months which were not apparent in recordings taken at 1 month of age. We have previously demonstrated that a developmental 'knockout' of the MNTB impacts ABR Wave III (*Jalabi et al., 2013*), so the decrease in Wave IV reflects changes in downstream areas of the auditory

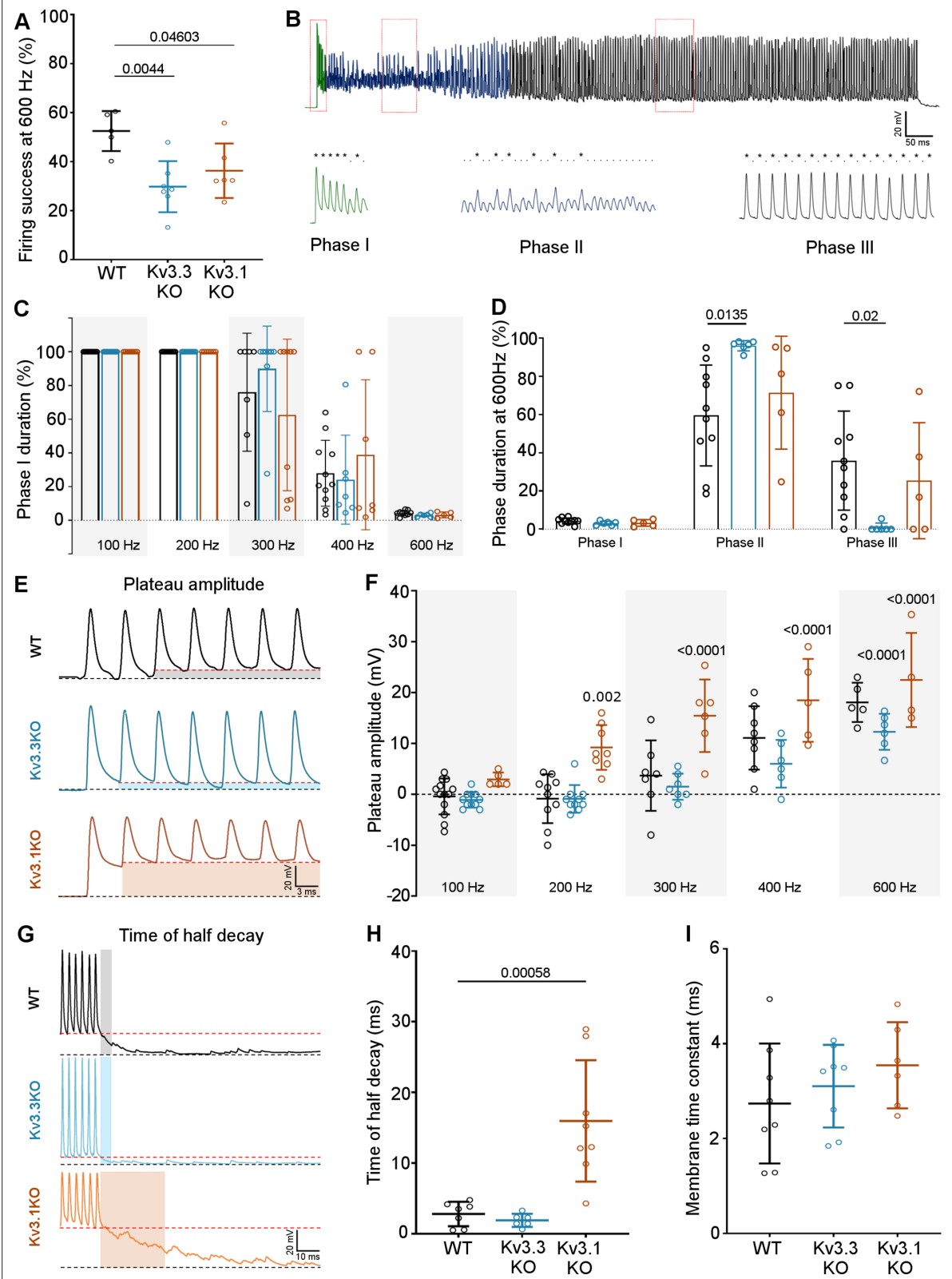

**Figure 7.** Kv3.3 deletion reduced ability to sustain MNTB AP firing at high frequencies. (**A**) Percent firing success of MNTB neurons for 600 Hz calyceal stimulation is reduced for both Kv3.3KO and Kv3.1KO.(**B**) A representative AP train recorded for an MNTB neuron in response to 600 Hz synaptic stimulation lasting 800ms. Three phases of input:output firing defined: Phase I (green) 1:1 AP firing calyx:MNTB for each EPSC is prevalent early in the train. Phase II (dark blue) follows phase I where some EPSCs drop below threshold, and AP firing becomes less probable and chaotic. Phase III (black)

*Figure 7 continued*

MNTB AP firing becomes regular again but is now firing to alternate input EPSCs, restoring information transmission. Expanded sections (boxed) of each phase of AP firing are shown below the full trace, APs are indicated by '*' and EPSPs that are below threshold are indicated by '.'. (**C**) Phase I duration across stimulation frequencies (100–600 Hz) for each genotype (WT, black; Kv3.3KO, blue; Kv3.1KO, orange): Phase I firing lasts throughout the train at frequencies up to 300 Hz but declines to only a few milliseconds at 600 Hz synaptic stimulation, but there were no significant differences between the genotypes. (**D**) The time spent in each phase for 600 Hz stimulation train (with each genotype indicated by the same colours as in C). The MNTB neuron is unable to maintain Phase I and Phase II dominates for each genotype. However, Phase III is only achieved briefly on 1 of 6 observations in the Kv3.3KO.(**E**) A sustained depolarized plateau was also observed, in the MNTB AP trains, as indicated by the shaded regions in this data from 300 Hz, and was particularly large the Kv3.1KO. (**F**) The amplitude of the depolarisation plateau increased in magnitude with stimulation frequency for each genotype, but was significantly larger in the Kv3.1KO at all frequencies above 100 Hz. (**G**) A slowly decaying depolarisation following the end of the synaptic train, as shown for representative traces from each genotype. (**H**) This decaying depolarisation is quantified as the time to half-decay and was significantly longer in the Kv3.1KO compared to WT in a 300 Hz AP train. (**I**) The postsynaptic MNTB neuron membrane time constant was unchanged across all genotypes (genotype: WT, black; Kv3.3KO, blue; Kv3.1KO, orange).

The online version of this article includes the following source data for figure 7:

**Source data 1.** Relates to *Figure 7*.

brainstem and is being characterized as part of another study. Here, we used the opportunity to examine auditory processing in the MNTB in vivo using extracellular recordings from WT and Kv3.3KO mice during sound stimulation. This data was non-gaussian and summary statistics are presented as median, and quartile values (in square brackets).

Extracellular MNTB single unit recordings exhibited a typical complex waveform, comprised of a presynaptic and a postsynaptic component (*Figure 8A*; *Kopp-Scheinpflug et al., 2003*). The time between the peak and trough of extracellular APs is a compelling marker for AP halfwidth (*Ritzau-Jost et al., 2021*) and confirmed our results of the presynaptic patch clamp recordings. AP halfwidth of the presynaptic AP (preAP) was significantly longer in Kv3.3KOs (0.25ms [0.16; 0.31]; n=13) compared to WT recordings (0.17ms [0.16; 0.19]; n=20; *Figure 8B*; Mann-Whitney Rank Sum Test: p=0.036). Synaptic delays as measured by peak-to-peak times in the complex waveform were also significantly prolonged in Kv3.3KOs (0.56ms [0.45; 0.70]; n=13) compared to WT controls (0.44ms [0.41; 0.49]; n=20; *Figure 8C*; n=13; Mann-Whitney Rank Sum Test: p=0.013). While changes in presynaptic AP duration and synaptic delay may predominantly affect temporal processing, the prolonged postsynaptic AP duration observed in the Kv3.3KOs (0.64ms [0.29; 0.46]; n=13; *Figure 8D*) might influence high-frequency firing abilities (WT: 0.36ms [0.48; 0.82]; n=20; Mann-Whitney Rank Sum Test: p≤0.001).

Temporal processing was tested by presenting suprathreshold sound stimuli at the neurons' characteristic frequency in 6–8 months old mice and comparing WT and Kv3.3KO strains. MNTB neurons in WT responded with a phasic-tonic firing pattern (*Figure 8E and F*) with short latencies (3.44ms [3.01; 4.21]; n=25; *Figure 8I and J*) and minimal jitter (0.18ms [0.09; 0.51]; n=18; *Figure 8I and K*). In contrast, Kv3.3KO neurons were slower to respond to sound with first spike latencies of 4.06ms [3.01; 4.21] (n=25; Mann-Whitney Rank Sum Test: p=0.015; *Figure 8I and J*) and showed larger temporal variability (0.92ms [0.56; 1.50]; n=18; Mann-Whitney Rank Sum Test: p=0.001; *Figure 8I and K*). This increased onset jitter contributed to the reduced magnitude of the phasic component in the peristimulus time histogram in Kv3.3KO neurons (*Figure 8F* arrows). The inability to fire high instantaneous rates at sound onset was accompanied by a shift in the average inter-spike intervals from 2.16ms [1.48; 3.57] (n=22) in the WT to 3.10ms [2.49; 4.19] (n=18) in the Kv3.3KO (Mann-Whitney Rank Sum Test: p≤0.001; *Figure 8G*). Indeed, comparing peak firing rates during the first 10ms of the sound-evoked response revealed significantly lower rates in Kv3.3KOs (240 Hz [180; 280]; n=18) compared to WTs (350 Hz [200; 425]; n=22; Mann-Whitney Rank Sum Test: p=0.003; *Figure 8H*). In contrast to the reduced sound-evoked firing rates, an increase in spontaneous firing was observed in the Kv3.3KOs (Kv3.3KO: 40 Hz [19.3; 62.0]; n=19) compared to WT controls (16.5 Hz [4.5; 42.8]; n=24; Mann-Whitney Rank Sum Test: p=0.044; *Figure 8L*). Together, these combined changes caused a significant reduction in signal-to-noise ratio in the Kv3.3KO (2.77 [2.04; 3.99]; n=14) compared to WT controls (9.70 [4.38; 40.95]; n=18; Mann-Whitney Rank Sum Test: p=0.001; *Figure 8M*).

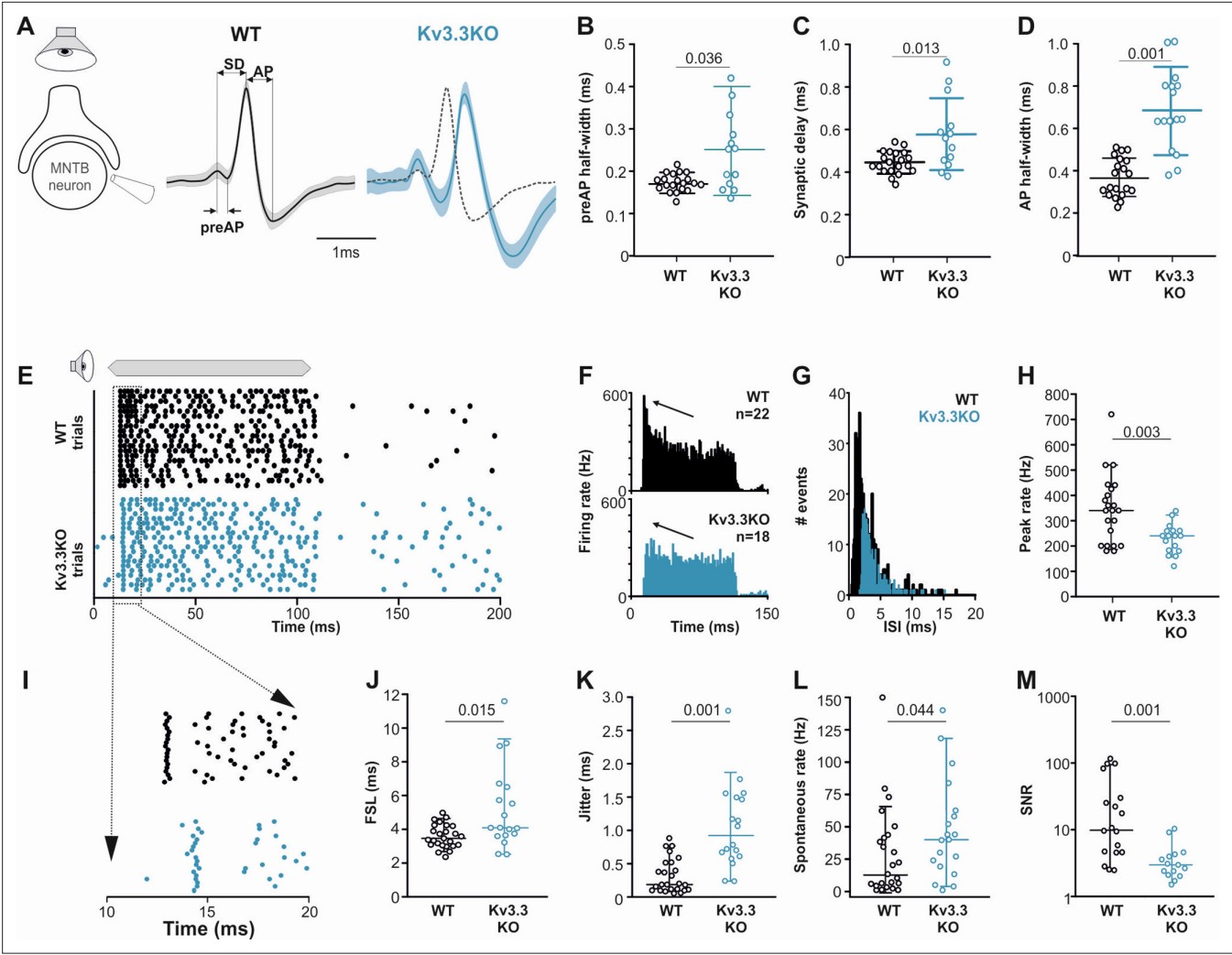

**Figure 8.** Presynaptic Kv3.3 accelerates the brainstem response to sound and improves timing and signal-to-noise ratio. (**A**) Extracellular recording from Calyx/MNTB in vivo shows complex APs (presynaptic and postsynaptic) from WT (black) and Kv3.3KO (blue) mice in response to sound; overlay of APs (right) shows delayed and longer APs in the KO. (**B, C, D**) Presynaptic AP half-width, synaptic delay and postsynaptic AP half-width are all longer in the Kv3.3KO (blue) than WT (black).(**E**) Raster display of MNTB AP response to sound (20 trials, 100ms duration) and spontaneous firing for both WT (black, upper) and Kv3.3KO (blue, lower). (**F, G, H**) Peri Stimulus Time Histogram (PSTH) of the evoked APs (grand avg. over all neurons, 1ms bins) show reduced peak firing rates (F, black arrow) due to longer interspike intervals (ISIs) and increased jitter (**I, K**) in the Kv3.3KO (blue) and the peak (0–10ms of the response) firing rate is significantly reduced. (**I, J, K**) Expansion of first 20ms of the Raster plot shows increased first spike latency and jitter (latency SD) in the Kv3.3KO (blue). (**L, M**) The mean spontaneous firing rate was higher in the Kv3.3KO (blue) and overall, these changes degraded signal-to-noise ratio in Kv3.3KO (blue) relative to WT (black). Data is presented as median and inter-quartiles. p Values calculated using Mann-Whitney Rank Sum Test and statistically significant values displayed on each graph.

The online version of this article includes the following source data for figure 8:

**Source data 1.** Relates to *Figure 8*.

## Discussion

Neurotransmitter release is triggered by calcium influx through voltage-gated calcium channels and is highly influenced by the presynaptic AP waveform. Deletion of Kv3.3 subunits increased presynaptic AP duration and transmitter release, enhancing short-term depression on repetitive stimulation. In contrast, deletion of Kv3.1 had little effect on the presynaptic AP or transmitter release. Expansion microscopy was used to enhance resolution in imaging presynaptic compartments and showed localisation of Kv3.3 subunits to the presynaptic membrane. These observations are consistent with a computational model of transmitter release in which Kv3.3 deletion increased vesicle release probability by twofold and to a lesser degree accelerated fast vesicle replenishment. This is consistent with

increased calcium influx and activity-dependent facilitation of recycling. The enhanced short-term depression of synaptic responses increased latency jitter and reduced postsynaptic AP firing (output) at high frequencies in the Kv3.3KO. In vivo this manifests as reduced temporal fidelity and firing rates in the binaural auditory circuit in response to sound, whilst spontaneous AP firing increased in the Kv3.3KO. We conclude that presynaptic Kv3 channels require one (or more) Kv3.3 subunits to achieve presynaptic targeting for fast and temporally precise transmitter release at this excitatory synapse.

On deletion of Kv3.3, synaptic transmission at the calyx of Held showed accelerated recovery from short-term depression, along with increased presynaptic AP duration and increased transmitter release from WT mice of a similar age. The magnitude and fast kinetics of Kv3 channels guarantee fast repolarisation and generate a fast afterhyperpolarisation (*Brew and Forsythe, 1995*; *Figure 1B*, here) which in concert with a resurgent Na$^+$ current (*Kim et al., 2010*; *Lewis and Raman, 2014*) maximizes the availability of voltage-gated Na$^+$ channels to maintain short APs during sustained firing, as also observed in vivo (*Sierksma and Borst, 2017*). Our results on transmitter release are consistent with enhanced activity-dependent vesicle recycling in the absence of Kv3.3, as a consequence of the increased AP duration and calcium influx (*Neher and Sakaba, 2008*; *Yang et al., 2014*; *Lipstein et al., 2021*). Our modelling indicates that the increase in transmitter release is primarily an increase in vesicle release probability (and subsequent short-term depression) but an acceleration of vesicle recycling also contributes.

## Changes in auditory processing over development

A key design element of this study was that each experiment compared mice of similar age against the three genotypes (WT, Kv3.3KO, and Kv3.1KO) all of which were on the same CBA background strain. Thus, variance contributed by the developmental stage was minimized within each experiment and the significant changes therefore related to the absence or presence of the Kv3 subunits under study. The range of animal ages used in this study (in different experiments) shows that Kv3.3 is relevant across development. Nevertheless, auditory responses are maturing, particularly around the onset of acoustically evoked activity (P12) in the mouse. The calyx of Held undergoes a range of developmental changes that foster temporally accurate transmission (*Borst and Soria van Hoeve, 2012*). Presynaptic calcium channels shift from N to P/Q type (*Iwasaki et al., 2000*), myelination induces developmental and activity-dependent changes (*Kim et al., 2013*; *Sinclair et al., 2017*) and similar adaptations favouring high-frequency firing are occurring in neonatal rat pups (*Sierksma and Borst, 2017*). The AP time-course accelerates (*Taschenberger and von Gersdorff, 2000*) with increased expression of Nav1.6 (*Leão et al., 2005*) which in turn influences activation of the voltage-gated calcium channels (*Borst and Sakmann, 1998*) and hence the efficacy of transmitter release (*Yang and Wang, 2006*). Presynaptic Kv currents also increase (*Nakamura and Takahashi, 2007*) with the major change being before hearing onset (and little change from P13 to P20). At the post-synaptic membrane, a concurrent switch to fast time-course GluA4-flop dominated AMPAR-mediated EPSCs (*Barnes-Davies and Forsythe, 1995*; *Geiger et al., 1995*; *Yang et al., 2011*) with developmental refinement in GluA2, reducing calcium permeability (*Lujan et al., 2019*) and a decrease in AMPAR desensitisation, further enhance transmission fidelity (*Taschenberger et al., 2002*; *Wong et al., 2003*). The present study interrogates the role of Kv3 subunits over a wide age range, from a relatively immature calyx (P10-P12) to mature adult (6 months old) and despite the inevitable developmental changes during this period, Kv3.3 remains crucial in shaping the presynaptic AP waveform and neurotransmitter release at the calyx. An increase in Kv3 expression with age may actually amplify the deficits observed in Kv3.3KOs at later developmental stages however this study shows that even with a small deficiency in Kv3.3 expression in young Kv3.3KO animals, there are drastic consequences at the synapse.

## Kv3 channel subunit composition

Knowledge of Kv channel subunit composition, interactions and precise location within identified neurons is key to understanding their extensive physiological roles in controlling neuronal excitability (*Trimmer, 2015*). Kv channel building is regulated by the N-terminal tetramerisation domain (T1) (*Li et al., 1992*) to favour assembly of 'dimers of dimers' (*Tu and Deutsch, 1999*) usually from subunits within the same Kv family. Although early studies suggested otherwise, Kv3 subunits do not co-assemble with the accessory subunit gene families (Kv5, Kv6, Kv8, and Kv9) (*Bocksteins et al., 2014*), so Kv3 channels are likely composed of four Kv3 alpha subunits. Frequent co-expression of Kv3

subunits in the same neurons suggest functional channels could exist as heteromers, indeed coimmunoprecipitation has revealed interactions of Kv3.1b with Kv3.4a and Kv3.2 in globus pallidus neurons (*Baranauskas et al., 2003*; *Hernández-Pineda et al., 1999*), and with Kv3.3 in the cerebellum (*Chang et al., 2007*) but homomeric assemblies of Kv3.3 subunits could also form presynaptic Kv3 channels. The data reported here suggests that presynaptic Kv3 channels at the calyx of Held require an obligatory Kv3.3 subunit, since TEA (which will block all Kv3 heteromers) showed little effect on transmission in the Kv3.3KO. This result also implies that Kv3.1 does not have a directive or independent role in the terminal and while heteromers with Kv3.3 may occur, Kv3.1 is certainly not required for Kv3.3 localisation to the presynaptic membrane.

## Could other Kv3 subunits contribute to presynaptic Kv3 channels here at the calyx or at other synapses?

Excitatory synaptic boutons, including hippocampal mossy fiber terminals (*Alle et al., 2011*), cerebellar mossy fiber terminals (*Ritzau-Jost et al., 2014*) and small neocortical boutons (*Ritzau-Jost et al., 2021*) show pharmacological evidence for presynaptic Kv3. Inhibitory interneurons also possess presynaptic Kv3 channels, including cerebellar basket cell terminals (*Southan and Robertson, 2000*) and stellate cell axons (*Rowan et al., 2016*), with the latter being identified as Kv3.4. However, it is difficult to exclude a role for Kv3.3 at this site since the Kv3 antagonist BDS is not specific for Kv3.4, having similar actions on Kv3.1 and Kv3.2; and should also block Kv3.3 subunits on the basis of sequence homology in the putative BDS binding site (*Yeung et al., 2005*). It seems unlikely that Kv3.4 (or Kv3.2) subunits have a directive (or independent) role in trafficking Kv3 channels to the calyx terminal, since the mRNA levels for these subunits was very low in the MNTB and VCN. However, if there were effective translation and assembly of Kv3.4 (or Kv3.2) into calyx Kv3 channels, then TEA should have enhanced transmitter release in the Kv3.3KO, but TEA potentiation of transmission was occluded in the Kv3.3KO.

In contrast to Kv3.1 and Kv3.2, early studies of recombinant Kv3.3 showed N-type (ball-peptide) inactivation, as originally described for Kv1 potassium channels (*Hoshi et al., 1990*). We conclude here that Kv3.3 is a principal component of presynaptic Kv3 channels but we observe little or no inactivation of the presynaptic Kv3 current. There are multiple explanations for this lack of inactivation. The degree of Kv3 inactivation varies between different expression systems (*Rudy et al., 1999*), perhaps due to there being two methionine start codons. Mammals lack a well-defined Kozak consensus sequence for the first, favouring initiation at the second methionine, thereby excluding the N-terminal peptide (*Fernandez et al., 2003*). Alternatively, endogenous PKC phosphorylation of N-terminal Kv3.3 could occlude the N-type inactivation (*Desai et al., 2008*). Also, little or no inactivation was observed in the postsynaptic Kv3.3 channel (*Choudhury et al., 2020*). Given this lack of inactivating current observed at the calyx, it is unlikely that Kv3.4 subunits are present in this terminal. Further to this, in recombinant co-expression studies, Kv3.3 and Kv3.4 did not readily form heteromers, but instead formed distinct currents with differing kinetics in the same cells (Richardson, Forsythe and Pilati, unpublished observations) suggesting that heteromeric channels composed of these subunits are unlikely.

In terminals that exhibit inactivation of presynaptic potassium currents, this has important physiological implications for transmitter release, especially during sustained high-frequency AP firing. Repetitive activation during AP trains would cause cumulative Kv3 inactivation and progressively slows AP repolarisation. This can generate a form of activity-dependent short-term potentiation where increasing AP duration potentiates transmitter release, as occurs at mossy fiber terminals (*Geiger and Jonas, 2000*). During sustained high-frequency AP firing at the calyx, the longer AP duration induced by an inactivating Kv3 would risk earlier transmission failure, with faster depletion of vesicles and inactivation of the presynaptic sodium channels. The physiological end-point for an inactivating Kv3 would be similar to that measured for the Kv3.3KO. Hence, the minimal inactivation of the presynaptic Kv3 current at the calyx of Held enables transmission to be sustained for longer, at higher firing rates and greater accuracy.

## Do interactions with the cytoskeleton dictate Kv3 localisation or function?

Proximity of $K^+$ channels have powerful influences on the local voltage and therefore in control of other voltage-gated ion channels. The data reported here demonstrates an ionic role for Kv3.3 in

the presynaptic terminal but stabilisation in any compartment will require non-ionic interactions with components of the cytoskeleton. The permutations and complexity of channel subunit assembly optimize function through subunit-specific localisation (*Trimmer, 2015*) involving trafficking, insertion, phosphorylation, and stabilisation of channel complexes. Indeed, a general scaffolding interaction is essential for intrinsic plasticity where insertion into key sites of excitability (axon initial segments, nodes of Ranvier, heminodes and synaptic terminals) will control hyper-excitability (*Steinert et al., 2011*) and calcium overload. There is a well-established role for ankyrin-G in axonal targeting of Kv3.1b (*Xu et al., 2007*) and for clustering of sodium and potassium channels at the initial segment (*Pan et al., 2006*). Ankyrin-R binds and stabilizes Kv3.1b in fast spiking interneuron somatic membranes and nodes of Ranvier; and crosslinks Kv3.3 via spectrin to the cytoskeleton in Purkinje neurons, which aids their survival (*Stevens et al., 2021*; *Stevens et al., 2022*). This interaction of both subunits with ankyrin-R would explain the inability of the MNTB neuron soma to differentiate or partition channels composed of Kv3.1b or Kv3.3 subunits (*Choudhury et al., 2020*) so that each subunit type could largely compensate for the absence of the other within the soma. Presynaptic stabilisation of Kv3.3 however, is unlikely to be through ankyrin-R alone, as this mechanism would not discriminate between the subunits.

Another possibility is that Kv3.3 inactivation is suppressed by the C-terminus associating with Hax1 (*Blosa et al., 2015*) to bind the Kv3.3 N-Terminal ball peptide and modulate the actin network. However, as in the WT, the calyceal outward currents reported in *Zhang et al., 2016* were an order of magnitude smaller than observed here or elsewhere. Additionally, the presynaptic K$^+$ currents from a Kv3.3KO calyx were similar magnitude to WT (Figure S4: *Zhang et al., 2016*). Based on association with the actin cytoskeleton, deletion of Kv3.3 was proposed to inhibit endocytosis in the calyx synaptic terminal by this non-ionic mechanism (*Wu et al., 2021*). However, the *Wu et al., 2021* study does not test for the possibility of a simultaneous ionic mechanism or for a change in the presynaptic AP waveform on deletion or mutation of Kv3.3. With the evidence reported here, it is clear that Kv3.3 does have an ionic role at the calyx of Held. Our observations on EPSCs and modelling of short-term depression show that Kv3.3 deletion increases transmitter release and accelerates a fast sub-second component of activity-dependent recycling (*Figure 5F*, inset) following conditioning stimulation. Although this contrasts with the slowing of endocytosis observed in *Wu et al., 2021* our observations are over a much shorter time-course and measure EPSC amplitude, rather than presynaptic capacitance. Future experiments will determine what part of the endocytosis capacitive changes result from direct or indirect ionic mechanisms versus non-ionic mechanisms.

## Relevance of Kv3.3 to mechanisms of sound localisation

The kinetics of Kv3 channels is fast enough to generate a rapid afterhyperpolarisation following the AP, which maximises the driving force for calcium influx and promotes recovery of voltage-gated sodium and calcium channels from inactivation. Short duration APs permit high firing rates by minimising the absolute refractory period, thus Kv3 enables well-timed high-frequency AP trains and when input firing rates exceed the refractory limit, neurons expressing Kv3 readily switch to firing on alternate input spikes (*Song et al., 2005*). In this study, we evoked MNTB neuron AP firing in response to presynaptic stimulation and asked how effective was the input/output relationship across a range of firing frequencies from 100 to 600 Hz in each genotype. Although there was little difference at low firing frequencies, at the highest frequencies only the first few presynaptic APs generate postsynaptic APs with 1:1 fidelity (Phase I) before entering a chaotic phase (Phase II) with many failures, which resolves later in the train into precise firing but to alternate stimuli (Phase III). In the absence of Kv3.3 the MNTB firing passed into Phase II but did not converge into Phase III firing, consistent with the idea that presynaptic Kv3.3 improved stability of synaptic transmission.

Integration of information from both ears for the purpose of sound localisation requires high temporal accuracy for bilateral transmission of auditory APs from each cochlea to the superior olivary complex (*Tollin, 2003*; *Joris and Trussell, 2018*). This is facilitated by Kv3 channels within specific neurons, from cell bodies, axons, synaptic terminals and across the neural network. In vivo extracellular recording from the MNTB measures simultaneously the presynaptic calyx and postsynaptic MNTB single unit APs. Accurate transmission of AP timing was compromised in the Kv3.3KO, and the peak firing rate was reduced. Additionally, spontaneous AP firing was elevated (with respect to WT animals), likely reflecting hyper-excitability upstream in the auditory pathway, with Kv3.3 being

expressed in spiral ganglion (*Kim et al., 2020*) and the globular bushy cells (*Li et al., 2001*; *Cao et al., 2007*) which give rise to the calyx of Held. The use of anesthetic for our in vivo study may affect auditory processing, but comparison with multiple other studies in mice and gerbil and using a range of different anesthetics showed that the synaptic delay measured in this study was favourably comparable to these other studies: ketamine/xylazine in mouse: 0.40ms (*Blosa et al., 2015*), 0.46ms (*Kopp-Scheinpflug et al., 2003*), 0.50ms (*Lorteije et al., 2009*) ketamine/xylazine in gerbil: 0.44ms (*Tolnai et al., 2009*). This narrow distribution of synaptic delays with different anesthesia suggests that the effect of anesthesia on synaptic delay is negligible compared to the effect of deleting Kv3.3.

## Kv3.3 physiology, dysfunction, and disease

Immunohistochemical studies have localized Kv3.3 subunits to many different synaptic terminals across the brain, from spiral ganglion afferent processes (*Kim et al., 2020*), medial vestibular nuclei (*Brooke et al., 2010*), cerebellar dentate nucleus (*Alonso-Espinaco et al., 2008*), parallel fiber synapses (*Puente et al., 2010*), posterior thalamic nucleus (*Chang et al., 2007*), and the neuromuscular junction (*Brooke et al., 2004*). It is interesting to consider why Kv3.3 is present at some synaptic terminals, but not all. One hypothesis is that presynaptic Kv3.3 would enhance resource and energy conservation during high frequency firing, but more importantly it will support the physiological mechanisms as a synapse evolves from a microdomain to a nanodomain architecture for exocytosis (*Meinrenken et al., 2003*; *Young and Veeraraghavan, 2021*) by constraining presynaptic calcium influx.

Disease mutations associated with Kv3.3, such as spinocerebellar ataxia type 13 (SCA13) cause multiple neurological defects, which from the evidence reported here could include aberrant neurotransmitter release in addition to postsynaptic excitability changes. In the case of the SCA13 mutation R420H (*Middlebrooks et al., 2013*), the observations made here support the hypothesis that SCA13 disrupts the physiological process of interaural level discrimination by reducing the gain and signal to noise ratio, thereby undermining sound source localisation (*Tollin, 2003*). The current study suggests several areas for further investigation of presynaptic potassium conductances: First, the contribution of ionic and non-ionic mechanisms is now recognized for Kv3 and their relative contribution to cytoskeleton and physiology can be elucidated. Second, it is necessary to determine which spliced-variants or potassium channel beta subunits/accessory proteins are expressed and how subunit phosphorylation contributes to the physiology. Third, while Kv3.3 can be clearly resolved on the non-release face of the terminal, the resolution on the release face is not sufficient to unambiguously distinguish between the pre- and post-synaptic membranes, but in order to avoid depolarisation by $[K^+]_o$ accumulation, one would predict minimal potassium efflux into the synaptic cleft. Fourth, Kv3 currents do not mediate all the presynaptic potassium current and other Kvs operating at more negative voltage activation ranges, such as Kv1 (*Geiger and Jonas, 2000*; *Dodson et al., 2003*) or Kv7 (*Zhang et al., 2022*) are likely to contribute to terminal threshold, excitability and AP waveform.

The combinatorial potential of Kv3 channel subunits gives rise to a spectrum of physiological roles in fast-spiking neurons (*Choudhury et al., 2020*) and interneurons (*Chang et al., 2007*). The pre- and postsynaptic studies conducted here at the calyx of Held/MNTB neuron clearly show that Kv3.3 subunits are critical for presynaptic localisation of Kv3 channels mediating fast AP repolarisation. They contribute to improved temporal accuracy and conservation of presynaptic resources during high-frequency firing. Together these results suggest a general role for Kv3.3 in raising the biophysical 'speed limit' for information transmission at fast synapses, especially during intense synaptic activity.

## Materials and methods

### Key resources table

| Reagent type (species) or resource | Designation | Source or reference | Identifiers | Additional information |
|---|---|---|---|---|
| Genetic reagent (*M. musculus*) | CBA/Crl | Charles River | Strain code: 609 | Leicester breeding colony |

*Continued on next page*

*Continued*

| Reagent type (species) or resource | Designation | Source or reference | Identifiers | Additional information |
|---|---|---|---|---|
| Genetic reagent (*M. musculus*) | Kv3.1 knockout mouse on CBA back-ground (Kv3.1KO) | Ho et al, PNAS 94, 1533–1,538. 1997. | CBA/CaCrl.*Kcnc1*<sup>tm1Joho</sup>/UoL | Strain originally created on 129/SV background and backcrossed onto CBA for >10 generations at the Preclinical Research Facility – University of Leicester |
| Genetic reagent (*M. musculus*) | Kv3.3 knockout mouse on CBA back-ground (Kv3.3KO) | Espinosa et al., J Neurosci 21, 6657–6,665. 2001 | CBA/CaCrl. *Kcnc3*<sup>tm1Echa</sup>/UoL | Strain was originally created on C57BL/6 background and backcrossed onto CBA for >10 generations at the Preclinical Research Facility – University of Leicester |
| Commercial assay or kit | Illumina NextSeq500 High Output | Illumina | FC-404–2002 | Used for Cochlear Nucleus |
| Chemical compound, drug | Strychnine hydrochloride | Sigma Aldrich (MERCK) | S8753-25G | (0.5 µM) |
| Chemical compound, drug | Tetraethylammonium | Sigma Aldrich (MERCK) | 86614–25 G | (1 mM) |
| Chemical compound, drug | Fentanyl | Janssen | 6,001 | (0.05 mg/kg) |
| Chemical compound, drug | Midazolam | B Braun | 17206034 | (5.0 mg/kg) |
| Chemical compound, drug | Medetomidine | Vetoquil GmbH | Domitor | (0.5 mg/kg) |
| Antibody | Anti-Kv3.1b (rabbit polyclonal) | Alomone | APC-014 | (1:1000) |
| Antibody | Anti-Kv3.3 (mouse monoclonal) | Neuromab | 75–354 | (1:3000) |
| Antibody | Alexafluor488 Goat anti-rabbit (Goat polyclonal) | Thermofisher | A-11008 | (1:1000) |
| Antibody | Alexafluor546 Goat anti-mouse (Goat polyclonal) | Thermofisher | A-11003 | (1:1000) |
| Antibody | Anti-Kv3.1b (rabbit polyclonal) | Synaptic systems | 242 003 (Lot# 1–2) | (1:300) |
| Antibody | Anti-Kv3.3 (rabbit polyclonal) | Alomone | APC-102 (Lot# APC 102AN0502) | (1:300) |
| Antibody | Anti-Bassoon (mouse monoclonal) | Synaptic systems | 141 111 (Lot# 1–2) | (1:300) |
| Antibody | Alexafluor488 Goat anti-mouse (goat polyclonal) | Invitrogen | A11001 (Lot# 21 40660) | (1:300) |
| Antibody | Alexafluor546 Goat anti-rabbit (goat polyclonal) | Invitrogen | A11010 (Lot# 21 89179) | (1:300) |
| Software, algorithm | Graphpad Prism 9.0.2 | Graphpad | RRID:SCR _002798 | |
| Software, algorithm | pClamp 10 software suite | Molecular Devices | RRID:SCR _011323 | |
| Software, algorithm | Zen Blue, 3.1 | Zeiss | RRID:SCR _013672 | |
| Software, algorithm | ABR Averager (Custom) | Wellcome Trust Sanger Institute | | |
| Software, algorithm | AudioSpike | HoerrTech | https://audiospike.hz-ol.de/ | |
| Software, algorithm | MATLAB R2010a | Mathworks | RRID:SCR_001622 | |

*Continued on next page*

*Continued*

| Reagent type (species) or resource | Designation | Source or reference | Identifiers | Additional information |
|---|---|---|---|---|
| Software, algorithm | Fiji | NIH ImageJ | RRID:SCR_002285 | |
| Other | Custom concentric bipolar stimulating electrodes | FHC | CBASD75 | |
| Other | Borosilicate glass capillaries | WPI | GC150F-7.5 | |

Experiments were conducted in accordance with the Animals (Scientific Procedures) Act UK 1986 and as revised by the European Directive 2010/63/EU on the protection of animals used for scientific purposes. All procedures were approved by national oversight bodies (UK Home Office, or Bavarian district government, ROB-55.2–2532.Vet_02-18-1183) and the local animal research ethics review committees.

Experiments were conducted on CBA/Crl mice (wildtype, WT) and knockouts were backcrossed for 10 generations onto this CBA/Crl background (*Choudhury et al., 2020*). PCR genotyping was done from ear notch samples made at P10. Mice were housed and breeding colonies maintained at the preclinical research facility (PRF) at the University of Leicester, subject to a normal 12 hr light/dark cycle and with free access to food and water (ad libitum). Both male and female animals were used in experiments with ages ranging from P10-25 for electrophysiology to 6 months of age for in vivo and auditory brainstem response (ABR) recordings.

## mRNA sequencing

Mice were killed by decapitation and brainstems were removed into 'RNA Later' stabilisation solution (Invitrogen, Cat# AM7020), before dissection to isolate both cochlear nuclei. Tissue from 3 CBA/Crl mice were pooled into 9 individual samples (27 mice total) before phenol extraction to isolate RNA. RNA purity, integrity, and concentration was assessed by UV-Vis spectroscopy (Nanodrop 8000) and capillary electrophoresis (Agilent Bioanalyzer 2000). Samples with RIN (RNA integrity)<7 were discarded. cDNA libraries were constructed using the NEBNext Ultra Directional RNA Library Prep Kit for Illumina sequencing performed using the Illumina NextSeq500 High Output (v2, 150 cycles) kit and the Illumina NextSeq500. Analysis of sequencing data was performed using Illumina Basespace. Using FastQC toolkit (Babraham Bioinformatics) total reads were trimmed of low-quality reads (Q<20), poly-A/T tails >10 bp, and adapter sequences before alignment to the mm10 (GRCM387) mouse genome (Ensembl) using TopHat2. Output values represented as Fragments per kilobase of transcript per million mapped reads (FPKM).

## Electrophysiology

### In vitro brain slice preparation

Mice were killed by decapitation, the brainstem removed and placed into ice-cold artificial cerebro-spinal fluid aCSF, oxygenated with 95%$O_2$/5%$CO_2$, containing (in mM): sucrose (250), KCl (2.5), $NaHCO_3$ (26), $NaH_2PO_4$ (1.25), D-Glucose (10), ascorbic acid (0.5) $MgCl_2$ (4), and $CaCl_2$ (0.1). For presynaptic recordings, 100-µm-thick transverse slices or 250-µm-thick slices for postsynaptic recordings were prepared in a pre-cooled chamber using a Leica VT1200S vibratome. Slices were allowed to recover for 1 hr at 37 °C in normal aCSF (*Choudhury et al., 2020*), continually bubbled with 95%$O_2$/5%$CO_2$ and subsequently allowed to passively cool to room temperature. The aCSF (310 mOsm) contained (in mM): NaCl (125), $NaHCO_3$ (26), D-Glucose (10), KCl (2.5), myo-inositol (3), $NaH_2PO_4$ (1.25), sodium pyruvate (1), ascorbic acid (0.5), $MgCl_2$ (1), and $CaCl_2$ (2). Recordings from brain slices were conducted at a temperature of 35±1°C.

### Presynaptic recordings

Mice aged P10-P12 were used for presynaptic calyx recordings. For each experiment, slices were placed in a recording chamber of a Nikon E600FN upright microscope and cells visualized with a 60 x DIC water-immersion objective (*Lucas et al., 2018*). Slices were continuously perfused with normal aCSF saturated with 95%$O_2$/5%$CO_2$, (as above), heated to 35 °C±1, at a rate of 1 ml/min. Whole-cell patch recordings were made using thick-walled borosilicate capillaries (1.5 mm OD, 0.86 mm ID) with

a resistance of 4–6 MΩ, filled with an internal solution composed of (in mM): KGluconate (97.5), KCl (32.5), HEPES (40), EGTA (0.2), MgCl$_2$ (1), K$_2$ATP (2.2), Na$_2$GTP (0.3), pH adjusted to 7.2 with KOH (295 mOsm). Calyces were identified visually, appearing as a second membrane profile around an MNTB principal neuron (*Billups and Forsythe, 2002a*; *Billups et al., 2002b*) using Differential Interference Contrast (DIC) optics. Presynaptic recordings were confirmed here through the total absence of spontaneous miniature synaptic events. (In 40 terminals 0 miniature synaptic events were observed in each 16 s calyceal recording (n=40) vs 10±7 events per second when recording from MNTB principal neuron; n=9, mice aged P10-P12). If any single spontaneous synaptic event was observed in a presynaptic recording, this data set was excluded. Additional criteria for calyceal recordings were that APs had a pronounced depolarising after-depolarisation potential (DAP, *Borst et al., 1995*; *Dodson et al., 2003*; *Kim et al., 2010*), a high membrane input resistance, a fast I$_H$ current (*Cuttle et al., 2001*) as opposed to a slow I$_H$ in the postsynaptic MNTB neuron (*Kopp-Scheinpflug et al., 2015*) and a resting membrane potential of around –70 mV, compared to –60 mV for postsynaptic neurons.

Recordings were made with a Multiclamp 700A amplifier (Molecular Devices), 1322A digidata (Axon Instruments) and pClamp 10 software (Molecular Devices) for acquisition and analysis. Electrode and cell capacitance were compensated and series resistances were corrected, with recordings discarded when series resistances reached >20 MΩ before compensation. All recordings were compensated by 70%. Signals were digitized at 100 kHz and filtered at 10 kHz. The stated voltages were not corrected for a liquid junction potential of 11 mV.

Current-voltage relationships were generated in voltage-clamp over a range of command voltages from –110 mV to +30 mV in 10 mV incremental steps. Steps were 150 ms in duration and separated by 1 s intervals. Action potentials measured in current-clamp were generated using short depolarising current steps (50 pA increments) of 50 ms duration, the first (threshold) evoked action potential was analyzed. Pipette capacitance was neutralized and the bridge balanced. Hyperpolarising current steps (–50 pA, 150 ms) were used to determine membrane resistance. For both voltage and current clamp experiments, voltage or current commands were from –70 mV.

## Postsynaptic recordings

The same experimental setup as described above was used for postsynaptic recordings. Borosilicate glass pipettes (2.5–3.5 MΩ resistance) were filled with a solution containing (in mM) KGluconate (120), KCl (10), HEPES (40), EGTA (0.2), MgCl$_2$ (1), K$_2$ATP (2.2). Electrode and cell capacitance were compensated and series resistances were corrected, with recordings discarded when series resistances reached >10 MΩ (before compensation) or changed by >10% during the recording. All recordings were compensated by 70%, signals were digitized at 100 kHz and filtered at 10 kHz.

Mice aged P20-P27 were used for studying synaptic physiology. Axons giving rise to the calyx terminal were stimulated using a concentric bipolar electrode (FHC, inc #CBAD75S) placed at the midline of a brainstem slice, controlled by a constant voltage stimulator box (DS2A, Digitimer) triggered by the pClamp 10 software. Axons were subjected to low-frequency stimulation (0.3 Hz) in order to determine the threshold for generating evoked excitatory postsynaptic currents (EPSCs) after which stimulation trains of 5 × 100, (separated by 20 s intervals), 200 and 600 Hz for 800ms separated by 30 s intervals were applied. Time was allowed for the internal patch solution to equilibrate (5 min) before stimulation trains were applied. EPSC recordings were conducted from a holding potential of –40 mV (to inactivate voltage-gated sodium channels) and inhibitory transmission blocked by adding 0.5 µM strychnine hydrochloride to external aCSF.

## Postsynaptic recordings of action potential trains at high frequencies

Current clamp recordings were made from MNTB neurons stimulated presynaptically with a bipolar concentric electrode (FHC, inc #CBAD75S). Trains of 0.1 ms stimuli of variable voltages (3–15 V) were delivered to the presynaptic axon at 100, 200, 300, 400, and 600 Hz using a constant voltage stimulation box (DS2A, Digitimer) triggered by the pClamp10 software. Evoked responses were recorded from the postsynaptic neuron under current clamp, with resting membrane potentials adjusted to –60 mV. Stimulation trains of the duration of 800 ms at the different frequencies were repeated 3 times and separated by 20 s intervals to allow for recovery.

## Voltage and current-clamp analysis

Electrophysiology analysis was conducted using Clampfit 10 software (Molecular Devices). Current amplitudes were measured as the steady-state current towards the end of the 150 ms voltage step. Presynaptic action potentials were analyzed using the threshold detection function. The threshold was set to the voltage of action potential activation and the relative amplitude defined as the difference between voltage at the peak and voltage at the threshold. Half-width is defined as the time delay between upstroke and downstroke at half-maximal amplitude. Rise and decay slopes were measured from 10% to 90% of peak amplitude.

Single excitatory postsynaptic currents were analyzed using the threshold detection function in pClamp. Baseline was defined as the resting current before stimulation. The threshold for detection was set to twice the standard deviation of the noise level. Peak amplitudes were defined as the difference between the current at the peak and the current at the baseline. Rise times were measured from 10% to 90% of the peak and decay taus were measured by fitting an exponential curve on the decay phase from 90% of the peak value. Charge was measured as the area under the curve, between the peak and baseline.

EPSC trains were analyzed by normalising each response to the first response in the train then fitting a single exponential to normalized amplitudes of responses and extracting the decay tau and steady-state amplitudes of the exponential fit to define the rate at which responses underwent short-term depression and the extent to which they depressed. mEPSCs were analyzed using template detection. A template of mEPSC kinetics was created by manually selecting 20 mEPSCs. This was then used for automatic detection of mEPSCs in all files.

## Immunofluorescence

Immunohistochemistry was conducted using similar methods to those reported in *Choudhury et al., 2020*. Briefly, mice aged 28 days were killed by decapitation and the brainstem dissected and immediately snap frozen in OCT before cutting at 12 µm transverse slices on a cryostat. Sections were mounted on poly-L-Lysine (Sigma Aldrich P8920) coated glass slides. For brainstem sections, tissues were removed and post-fixed with 4% PFA for 10 min at 4 °C. Antigen retrieval was achieved by incubating fixed tissues in a 10 mM Citrate buffer for 20 min at 85–90°C. For the cochlea both inner ears were dissected and fixed by perfusion at the round and oval window with 4% PFA in 1xPBS + 0.1% TritonX-100 (PBST), followed by overnight submersion in fixative at 4 °C. Cochlea sections were cut from whole tissues decalcified for 72 hr at 4 °C in 4% EDTA, cryoprotected with 30% sucrose, then embedded in OCT before sectioning at 12 µm on a cryostat and mounted on poly-L-Lysine (Sigma Aldrich P8920) glass slides. Antigen retrieval was achieved by 5 min incubation in 1% SDS prior to staining. Low-affinity binding was blocked using 10% Normal Goat Serum +1% BSA in PBST ('blocking solution') for 1 hr at room temperature. Afterwards primary antibodies for Kv3.1b (Rabbit, Alomone APC-014, 1:1000) and Kv3.3 (Mouse, Neuromab 75–354, 1:3000) were diluted in blocking solution and incubated overnight at 4 °C. Following 3 × 10 min washes in PBST, sections were incubated for 2 hours at room temperature with the appropriate secondary antibodies diluted 1:1000 in blocking solution (AF488 Goat Anti-Rabbit, A-11008; AF546, Goat Anti-Mouse, A-11003, Thermo Fisher). Slides were washed 3 × 20 min in PBST before cover slipping with hard-set mounting medium (Vectashield, H-1400) and stored at 4 °C. Slides were imaged using Leica DM 2500 and image analysis performed using Fiji software. The same observations were made in at least 3 animals for WT and 2 animals for each KOs.

## Protein retention expansion microscopy (proExM)

Mice aged P28-P30 were killed by decapitation, their brain dissected and brainstem sectioned to obtain transverse slices of 100 µm thickness. Tissue from three animals per genotype was used for each investigation. Free-floating brain slices underwent fixation (PBST, 4% PFA, 4 °C, 15 min), and then washed in PBST (3 × 10 min). A tissue blocking stage (RT, 1 h) was used to minimize low affinity binding with 1% BSA +10% Normal Goat Serum (Vector Laboratories S-1000) in PBST ('blocking solution'). Free floating slices were incubated with the primary antibodies in a petri dish: Anti-Kv3.1b rabbit (Synaptic Systems 242 003 Lot# 1–2, polyclonal), or Anti-Kv3.3 rabbit (Alomone APC-102 Lot# APC102AN0502, polyclonal), and Bassoon mouse Cl179H11 (Synaptic Systems 141 111 Lot# 1–2, monoclonal), diluted in blocking solution (1:300 for all primary Abs) and incubated overnight at room

temperature on a shaker. The slices were washed 3 × 10 min in PBST, and incubated with the respective secondary antibodies (AF488 goat anti-mouse IgG, 1:300 Invitrogen A11001, Lot# 2140660; AF546 goat anti-rabbit IgG, 1:300 Invitrogen A11010, Lot# 2189179) for 2 hr at room temperature on a shaker. Slices were washed 3 × 10 min in PBST and incubated with 0.1 mg/ml 6-((acryloyl)amino) hexanoic acid (AcX) solution in PBS overnight with no shaking, followed by 2 × 15 min wash in PBS. The slices were then 'gelled' and the gels digested according to the proExM protocol for intact tissues (basic protocol 2, *Asano et al., 2018*). The digested gels were expanded in ddH$_2$O water for 3 × 20 min for same-day imaging or stored at 4 °C in the dark in PBS for later use. For imaging, gels were transferred to 35 mm glass-bottom petri dishes with a sealed lid, previously coated with poly-L-Lysine (Sigma Aldrich P8920). Fluorescence imaging was conducted using a Zeiss LSM 980 Airyscan 2 microscope and images were processed using ZEN 3.1 (blue edition) and Fiji software. Controls for non-specific immunostaining of the secondary antibodies (incubations without the primary antibody, data not shown), and pre-incubation of each primary antibody with the corresponding antigenic peptide (blocking peptide) were conducted as part of the antibody validation procedure. Immunostaining with Kv3.1b and Kv3.3 primary antibodies was confirmed as absent in the corresponding knockout tissues.

## In vivo physiology

Adult (6–8 month) Kv3.3 knockout mice of either sex (n=5) and five age-matched CBA wild type mice were anesthetized with a subcutaneous injection of 0.01 ml/g MMF (0.5 mg/kg body weight Medeto-midine, 5.0 mg/kg body weight Midazolam and 0.05 mg/kg body weight Fentanyl). They were placed on a temperature-controlled heating pad (WPI: ATC 1000) in a soundproof chamber (Industrial Acoustics). Depth of anesthesia was measured using the toe pinch reflex and animals responding were given supplemental MMF at 1/3 the initial dose. The mice were stabilized in a custom stereotaxic device. An incision was made at the top of the skull, followed by a craniotomy just anterior to the lambda suture intersection. The skull was tilted to provide access to the auditory brainstem. A ground electrode was placed in the muscle at the base of the neck. Glass microelectrodes were pulled from glass capillaries so that the resistance was 5–20 MΩ when filled with 3 M KCl solution. Signals were amplified (AM Systems, Neuroprobe 1600), filtered (300–3000 Hz; Tucker-Davis-Technologies PC1) and recorded (~50 kHz sampling rate) with a Fireface UFX audio interface (RME). AudioSpike software (HoerrTech) was used to calibrate the multi-field magnetic speakers, generate stimuli and record action potentials. Stimuli consisted of pure tones (50–100 ms duration, 5ms rise/fall time) at varying intensity (0–90 dB SPL) and were presented through hollow ear bars connected to the speakers with Tygon tubing. PSTHs were assessed at characteristic frequency (CF) and 80 dB SPL. MNTB neurons were identified by their excitatory response to contralateral sound stimulation and their typical complex waveform (*Kopp-Scheinpflug et al., 2003*), consisting of a presynaptic potential (preAP), a synaptic delay (SD) and a postsynaptic potential (AP).

## Auditory-evoked brainstem response

ABR equipment set-up and recordings have previously been described in detail in *Ingham et al., 2011*. Briefly, mice were anesthetized with fentanyl (0.04 mg/kg), midazolam (4 mg/kg), and mede-tomidine (0.4 mg/kg) by intraperitoneal injection. Animals were placed on a heated mat inside a sound-attenuated chamber, and electrodes were inserted sub-dermally; below the right pinnae, into the muscle mass below the left ear, and at the cranial vertex. ABR responses were collected, amplified, and averaged using the TDT System3 (Tucker Davies Technology) in conjunction with custom 'Averager' software, provided by the Wellcome Trust Sanger Institute. Binaural stimuli were delivered in the form of a 0.1ms broadband click. All stimuli were presented in 5 dB SPL rising steps to 95 dB SPL, and responses were averaged 512 times per step. Recordings were averaged over a 20ms period with a 300–3000 Hz bandwidth filter and a gain of 25,000 x. Wave amplitude and latencies were analyzed using the Auditory Wave Analysis Python script developed by Bradley Buran (Eaton-Peabody Laboratory), and calculated as the difference between peak and valley (µV) and time to wave peak (s), respectively.

## Computation model

A simple model with activity-dependent vesicle recycling (*Graham et al., 2004*; *Billups et al., 2005*; *Lucas et al., 2018*) was used. In the model, vesicles in a releasable pool of normalized size *n* may

release with a fixed probability $P=P_v$ on the arrival of a presynaptic action potential at time $s$ to give an EPSC amplitude proportional to $np$ (equ. 3). Vesicles in this releasable pool are replenished up to the maximum normalised pool size of n=1 at a rate $\tau_r$ from an infinite reserve pool (equ. 1). In the absence of presynaptic action potentials, replenishment proceeds at a constant background rate (time constant $\tau_b$). Following a presynaptic action potential, the replenishment rate is instantaneously raised to a higher rate, $\tau_h$ (equ. 2b) which then decays back to the background rate with time constant $\tau_d$ (equ. 2 a). The model equations are:

$$\frac{dn}{dt} = \frac{1-n(t)}{\tau_r(t)} - \sum_s P_v . n(t) . \delta(t-s) \tag{1}$$

$$\frac{d\tau_r}{dt} = \frac{\tau_b - \tau_r(t)}{\tau_d} \tag{2a}$$

$$\tau_r(s) = \tau_h \tag{2b}$$

$$EPSC(s) = n(s) . P_v \tag{3}$$

The model is implemented in Matlab. Differential equations are solved by simple forward Euler integration.

## Statistics

Statistical analysis of the in vitro data was performed in Graphpad Prism V7 unless otherwise specified. Data were tested for a normal gaussian distribution using a Shapiro-Wilk normality test and parametric (one-way ANOVA) or non-parametric tests (Kruksal-Wallis ANOVA) applied as appropriate. Multiple comparisons were corrected for using Tukey's multiple comparisons test or Dunn's multiple comparison test post-hoc and a p value of<0.05 was taken as significant. The statistical tests applied are noted in the figure legends and corresponding text. Data is represented as mean ± SD unless otherwise stated. In vivo data are presented as medians and inter-quartiles in text numbers and figures in addition to individual data points. Statistical analyses of the in vivo data were performed with SigmaStat/SigmaPlot. Normality was tested by the Shapiro-Wilk Test. Comparisons between data sets were made using parametric tests for normally distributed data (two-tailed Student's t-test for comparing two groups) and when the normality assumption was violated, a non-parametric test (Mann-Whitney Rank Sum Test) was used.

## Acknowledgements

We are grateful to the Preclinical Research Facility at the University of Leicester for the animal care, husbandry and expert assistance provided, also to Neil Ingram for assistance and advice in setting up the ABR recording system. We thank the Advanced Imaging Facility (RRID:SCR_020967) at the University of Leicester for their support, including BBSRC funding (BB/S019510/1). This research was funded by a BBSRC project grant (R001154/1: VC, IDF) and a BBSRC Case PhD Studentship (M016501: AR, NP, IDF) including support from Autifony Therapeutics Ltd, and an EU H2020 LISTEN (722098) International Training Network postgraduate funding supporting KB (IDF). Further funding was provided by DFG SFB870 A-10 (CKS) supporting MS. Thanks also for the support provided to IDF by Benedikt Grothe during a sabbatical in the Division of Neurobiology, Faculty of Biology Ludwig Maximilian University, Munich, Germany.

## Additional information

### Competing interests

Nadia Pilati: This author is employed by Autifony Therapeutics Ltd. The other authors declare that no competing interests exist.

## Funding

| Funder | Grant reference number | Author |
|---|---|---|
| Biotechnology and Biological Sciences Research Council | R001154/1 | Ian Forsythe |
| Biotechnology and Biological Sciences Research Council | Case Award M016501 | Ian Forsythe |
| H2020 Health | ITN LISTEN 722098 | Ian Forsythe |
| Deutsche Forschungsgemeinschaft | DFG SFB870 A-10 | Conny Kopp-Scheinpflug |

The funders had no role in study design, data collection and interpretation, or the decision to submit the work for publication.

## Author contributions

Amy Richardson, Formal analysis, Investigation, Methodology, Writing – original draft, Writing – review and editing; Victoria Ciampani, Mihai Stancu, Sherylanne Newton, Bruce P Graham, Formal analysis, Investigation, Writing – review and editing; Kseniia Bondarenko, Formal analysis, Investigation, Methodology, Writing – review and editing; Joern R Steinert, Formal analysis, Investigation, Supervision, Writing – review and editing; Nadia Pilati, Funding acquisition, Investigation, Supervision, Writing – review and editing; Conny Kopp-Scheinpflug, Formal analysis, Funding acquisition, Investigation, Project administration, Supervision, Writing – review and editing; Ian D Forsythe, Conceptualization, Formal analysis, Funding acquisition, Investigation, Methodology, Project administration, Supervision, Writing – original draft, Writing – review and editing

## Author ORCIDs

Amy Richardson ⓘ http://orcid.org/0000-0002-1552-2915
Victoria Ciampani ⓘ http://orcid.org/0000-0002-4154-1562
Kseniia Bondarenko ⓘ http://orcid.org/0000-0003-3321-9423
Sherylanne Newton ⓘ http://orcid.org/0000-0002-8210-3526
Joern R Steinert ⓘ http://orcid.org/0000-0003-1640-0845
Ian D Forsythe ⓘ http://orcid.org/0000-0001-8216-0419

## Ethics

Experiments were conducted in accordance with the Animals (Scientific Procedures) Act UK 1986 and as revised by the European Directive 2010/63/EU on the protection of animals used for scientific purposes. All procedures were approved by national oversight bodies (UK Home Office, or Bavarian district government, ROB-55.2-2532.Vet_02-18-1183) and the local animal research ethics review committees. In vivo experiments were conducted under anaesthesia: with a subcutaneous injection of 0.01ml/g MMF (0.5mg/kg body weight Medetomidine, 5.0mg/kg body weight Midazolam and 0.05mg/kg body weight Fentanyl). Every effort was made to minimise suffering and at the end of each procedure the animal was humanely killed using an approved method.

## Decision letter and Author response

Decision letter https://doi.org/10.7554/eLife.75219.sa1
Author response https://doi.org/10.7554/eLife.75219.sa2

# Additional files

## Supplementary files

• Supplementary file 1. Summary table of averaged data and statistical tests presented in the figures, including significant (red) and non-significant values (black). Data are presented in the same order as figures appear in the article. Details are provided about the number of animals, number of cells, statistical test used and calculated p values.

• Transparent reporting form

## Data availability

Data generated in this study are included in the manuscript and supporting files. Source data files for each figure has been uploaded onto FigShare. Datasets Generated for the Ms "Kv3.3 subunits control presynaptic action potential waveform and neurotransmitter release at a central excitatory synapse" Authors: Ian D. Forsythe, Amy Richardson, Victoria Ciampani, Mihai Stancu, Kseniia Bondarenko, Sherylanne Newton, Joern Steinert, Nadia Pilati, Bruce Graham, Conny Kopp-Scheinpflug, 2022, https://figshare.com/s/9c0a07ed2fe5761cc281. The model code and associated data files are available at: Bruce Graham, 2021, https://github.com/bpgraham/CoH-Models, (copy archived at swh:1:rev:6ae468a42fc94dec2cf3f7c5490593ee321c8321).

The following datasets were generated:

| Author(s) | Year | Dataset title | Dataset URL | Database and Identifier |
|---|---|---|---|---|
| Forsythe ID, Richardson A, Ciampani V, Stancu M, Bondarenko K, Newton S, Steinert J, Pilati N, Graham B, Kopp-Scheinpflug C | 2022 | Kv3.3 subunits control presynaptic action potential waveform and neurotransmitter release at a central excitatory synapse | https://doi.org/10.25392/leicester.data.19322864.v1 | figshare, 10.25392/leicester.data.19322864.v1 |
| Graham B | 2021 | The model code and associated data files | https://github.com/bpgraham/CoH-Models | GitHub, bpgraham/CoH-Models |

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
