## [Editor Report]

This work shows that Kv3.3 potassium channels play a major role in shaping the presynaptic action potential waveform of calyx-type auditory synapses. Mice that lack Kv3.3 showed auditory response deficits, including increases of spike latency and jitter. Overall, the study shows the uniquely important role for Kv3.3 channels in the fast synaptic transmission between the neurons that compute sound localization in mammals.

---

## [Decision Letter]

**Decision letter after peer review:**

Thank you for submitting your article "Kv3.3 subunits control presynaptic action potential waveform and neurotransmitter release at a central excitatory synapse" for consideration by *eLife*. Your article has been reviewed by 3 peer reviewers, including Henrique von Gersdorff as the Reviewing Editor and Reviewer #1, and the evaluation has been overseen by Richard Aldrich as the Senior Editor.

The reviewers have discussed their reviews with one another, and the Reviewing Editor has drafted this to help you prepare a revised submission. Some further analysis of the data (mEPSC amplitudes) and discussion of the results is recommended.

Essential revisions:

1. It follows that the decay of the presynaptic AP slope in Kv3.3KO correlated with a broadening of the AP waveform. It is curious that the AP waveform did not broaden with a Kv3.1 KO, yet the decay slope slowed almost to the extent of the Kv3.3KO. Could it be that Kv3.1 is acting at slightly different time-points during repolarization? What about the AP width at 75% or 25% of the amplitude?

2. Figure 1B-C: the original study in JPhysiol (Forsythe, 1994) on the calyx AP shows that current step injections produce multiple APs in the presynaptic calyx, but just one in the postsynaptic MNTB cell. I thus wonder if maybe the recordings shown in Figure 1B-C of a solo AP may be perhaps from postsynaptic MNTB principal cells? Can the authors confirm with fluorescent dyes that the recordings shown in Figure 1B-C are indeed from the presynaptic calyx (e.g. they can mention no mEPSCs)? Important to mention also that mice start to hear at about P12, so P10-12 calyces are still immature.

3. in vitro slice recording was performed in mouse pups (P10-P25), whereas in vivo recording was performed in older mice (6 months). It is not clear why the experiments were done in mice with such huge age difference. The synapse might not be fully mature around P10-P25. Kv channels also tend to have increased expression during development. Were there any published results of immuno-electron microscopy to demonstrate the presence and expression levels of Kv3.3 and Kv3.1 channels in the calyx of Held terminals during these developmental stages?

4. There may be potential compensatory effects from other Kv3 subunits in Kv3.1 or Kv3.3 global KO mice. Are Kv3.3 channels expressed in the postsynaptic cell? Potential developmental changes caused by global KO should be more extensively discussed.

5. Figure 2: Could the increase in EPSC sizes be due to postsynaptic effects such as increased mEPSCs? This may be done by looking at asynchronous or spontaneous mEPSCs. In Figure 2, mini EPSCs should also be analyzed to help dissect the role of Kv3.3 at different stages of synaptic transmission.

6. The potential effect of anaesthesia on in vivo recording in response to sound should be discussed. The FSL difference in the Kv3.3 KO may arise from spiral ganglia and CN neurons firing more slowly. These neurons also may express Kv3.1 and Kv3.3. Please emphasize this in Discussion.

*Reviewer #1 (Recommendations for the authors):*

1. It follows that the decay of the presynaptic AP slope in Kv3.3KO correlated with a broadening of the AP waveform. It is curious that the AP waveform did not broaden with a Kv3.1 KO, yet the decay slope slowed almost to the extent of the Kv3.3KO. Could it be that Kv3.1 is acting at slightly different time-points during repolarization? What about the AP width at 75% or 25% of the amplitude?

2. Figure 1B-C: the original study in JPhysiol (Forsythe, 1994) on the calyx AP shows that current step injections produce multiple APs in the presynaptic calyx, but just one in the postsynaptic MNTB cell. I thus wonder if maybe the recordings shown in Figure 1B-C of a solo AP may be perhaps from postsynaptic MNTB principal cells? Can the authors confirm with fluorescent dyes that the recordings shown in Figure 1B-C are indeed from the presynaptic calyx (e.g. they should show no mEPSCs)? Important to mention also that mice start to hear at about P12, so P10-12 calyces are still immature.

3. Figure 6A: Kv3.1 KO show major effects in postsynaptic principal cell excitability and firing. This should perhaps be mentioned in the Abstract. Postsynaptic firing requires both Kv3.1 and Kv3.3.

*Reviewer #3 (Recommendations for the authors):*

I have a few points, which may be addressed by additional experiments and rewriting.

1) The increase EPSC sizes are not due to postsynaptic effects such as increased mEPSCs? This may be done by looking at asynchronous or spontaneous mEPSCs.

2) Kv3.3 channels are expressed in the postsynaptic cell, which may affect postsynaptic firing. One way will be to perform dynamic clamp in WT cells and apply synaptic conductance of WT and KO. Nevertheless, the authors may discuss this point carefully. Although it is likely that most effects are presynaptic, it is difficult for me to assess presynaptic affects precisely.

3) It is a bit difficult to understand why firing is reduced at the onset of sound stimulation in Kv3.3 KO mice. Overall, Fig7 is complex observation, and one needs some discussion.

4) Slice experiments and in vivo experiments use different ages. Is it reasonable to assume that the impact of Kv3.3 is the same?

---

## [Author Response]

Essential revisions:1. It follows that the decay of the presynaptic AP slope in Kv3.3KO correlated with a broadening of the AP waveform. It is curious that the AP waveform did not broaden with a Kv3.1 KO, yet the decay slope slowed almost to the extent of the Kv3.3KO. Could it be that Kv3.1 is acting at slightly different time-points during repolarization? What about the AP width at 75% or 25% of the amplitude?

Yes, this is an interesting observation and we have conducted further analysis of the presynaptic action potential to respond to this point. We see little difference between the Kv3.1KO and WT when measured near the peak (75%) or bottom (25%) of the AP, as shown in the new Figure 1—figure supplement 4. The small difference in the 25% and 75% AP widths from WT to Kv3.1KO was not significant at p = 0.05. We have added this data (Figure 1—figure supplement 4) and interpretation to the manuscript on page 4 (last paragraph). The significant decrease in decay slope in Kv3.1KO compared to WT is likely an artefact of the APs in Kv3.1KOs being slightly smaller than those observed in WT, amplifying the difference in the change of voltage over time. We are not excluding a role for Kv3.1-containing channels in influencing the presynaptic AP, since they are located in axonal nodes of Ranvier (See Devaux et al., 2003) and can form heteromers with Kv3.3 subunits in functional Kv3 channels, but we clearly show here that presynaptic Kv3 channels must contain at least one Kv3.3 subunit. We have added IHC of Kv3.3 and Kv3.1 in the calyx using expansion microscopy (New Figure 2); this confirms the dominance of Kv3.3 subunits in the terminal.

2. Figure 1B-C: the original study in JPhysiol (Forsythe, 1994) on the calyx AP shows that current step injections produce multiple APs in the presynaptic calyx, but just one in the postsynaptic MNTB cell. I thus wonder if maybe the recordings shown in Figure 1B-C of a solo AP may be perhaps from postsynaptic MNTB principal cells? Can the authors confirm with fluorescent dyes that the recordings shown in Figure 1B-C are indeed from the presynaptic calyx (e.g. they can mention no mEPSCs)? Important to mention also that mice start to hear at about P12, so P10-12 calyces are still immature.

The original 1994 paper which first demonstrated presynaptic recording from the calyx, included data over a broad age range of 6-12 days, and most groups which subsequently took up this preparation were using animals of 8-10 days old because of the difficulty in making these types of recordings. We showed that by 10 days old the calyx calcium channel compliment has largely switched to P-type, as noted in Forsythe et al., 1998 (p 798 & Figure 1) and documented in detail by Tomoyuki Takahashi with Osvaldo Uchitel in Iwasaki et al., 2000 (Figure 3). In 2003 we showed (Dodson et al., 2003) that block of presynaptic Kv1.2 channels caused multi-spiking in the calyx and Nakamura and Takahashi (2007) demonstrated the developmental acceleration in the calyx AP halfwidth (Figure 1) from 0.5ms at P7/8 to 0.25ms at P13/14 which was unchanged at P19/20. Comparing their AP halfwidth data at P13/14 and P19/20 with ours (0.28ms) suggests that our data is similar to their values from more mature animals. Nakamura and Takahashi also showed a developmental increase in calyceal Kv1 currents (Figure 8) from P7 to P14, which changed spiking from multiple AP firing to single spiking, confirming our observations for the role of Kv1 and specifically highlighting the developmental importance. Thus, our interpretation of the multi-spiking presynaptic recording from Forsythe 1994 is that these observations were from relatively immature animals lacking Kv1. A further change from many of the early studies on the calyx is the transition from rat to mouse as the experimental animal, which occurred as many groups in the field adapted to transgenic technology.

We have been very careful to exclude contamination of our presynaptic data with postsynaptic recordings. We have a long experience of making presynaptic recordings and although presynaptic fluorescent dyes can be helpful to identify the recording site, we try to avoid dye use when recording from the calyx in order to reduce phototoxicity and free radical damage. Miniature synaptic currents or potentials have never been observed in any calyceal recording (and no axo-axonic synapses have been reported on the calyx). Presynaptic recordings were confirmed empirically through the total absence of spontaneous miniature synaptic events; this is very reliable in our hands and none of these presynaptic recordings are mistakenly from a postsynaptic neuron. As evidence for this no mini events were observed in 16s recordings from each calyx (n=40 in total). In contrast, postsynaptic recordings showed an average of 10±7 mEPSCs per second (range 3-26 events n=9, mice aged P10-P12). In our criteria to confirm a presynaptic recording, if even 1 synaptic event was observed during the whole presynaptic recording, this data was excluded from the data set. Additional confidence for presynaptic recordings were that presynaptic APs had a pronounced depolarizing after-depolarisation potential (DAP) exhibited a fast I_H_ current and had a resting membrane potential of around -70mV, compared to -60mV for postsynaptic cells. A paragraph on this has been added to the methods . We have also added a paragraph to the discussion on development of the calyx that includes a statement about hearing onset.

3. in vitro slice recording was performed in mouse pups (P10-P25), whereas in vivo recording was performed in older mice (6 months). It is not clear why the experiments were done in mice with such huge age difference. The synapse might not be fully mature around P10-P25. Kv channels also tend to have increased expression during development. Were there any published results of immuno-electron microscopy to demonstrate the presence and expression levels of Kv3.3 and Kv3.1 channels in the calyx of Held terminals during these developmental stages?

It is important to appreciate that a key element of our experimental design was to compare the same age ranges against the three different genotypes (WT, Kv3.1KO and Kv3.3KO). Thus, the sources of variance contributed by development should have been similar and any statistically significant differences should be directly related to the absence or presence of the specific subunits (Kv3.3 or Kv3.1).

The range of ages used for this study was to provide as much information as possible to the community and to show that the role of presynaptic Kv3 contributes at all ages studied. Additionally, several of the techniques were hard to apply over the whole age-range. For instance, patch clamp recording is more challenging in the spinal cord and brainstem of animals over 1 month old, while in vivo recording is harder in younger animals.

An increase in channel subunit expression is likely to amplify differences between WT and Kv3.3KO at older time points, however this does not undermine the observations in the younger animals. The expansion microscopy that we have added shows that both Kv3.1b and Kv3.3 are present in the terminal of mice aged ~P20-P30. We have already published quantification of Kv3 mRNA from P9 to P30 in the MNTB and other nuclei in the auditory brainstem; this showed no significant developmental increase in mRNA over this age range, but a high intrinsic variance. We have repeated this study twice (as noted in Choudhury et al. 2020) and even with increased N, there was no statistically significant increase in mRNA levels over this age-range.

Nevertheless, the development of the calyx/MNTB neuron excitability and synaptic transmission are important and well documented, and we have discussed the evidence in greater detail by adding a paragraph in the discussion. The work of Nakamura and Takahashi (2007) is key data which clearly shows that while there is an increase in Kv currents with development, most of this change has occurred before the age of P13, as there isn’t a significant change in the AP width from P13-P20 in their WT study.

Many groups including ourselves have conducted developmental studies of synaptic transmission and excitability at the calyx of Held. We have added a paragraph discussing these developmental studies . We have also added discussion of Kv3 potassium channel subunits in EM studies , particularly relevant now we have conducted new experiments using Expansion Microscopy for presynaptic localisation of Kv3.3 and Kv3.1. See new results and data in new Figure 2.

4. There may be potential compensatory effects from other Kv3 subunits in Kv3.1 or Kv3.3 global KO mice. Are Kv3.3 channels expressed in the postsynaptic cell? Potential developmental changes caused by global KO should be more extensively discussed.

We have already excluded the possibility of compensatory changes by other Kv3 subunit mRNA in the knockouts used in this study (this is reported in Choudhury et al., 2020 cited in the first paragraph of the results). A detailed discussion of compensatory issues in the Kv3 knockouts was made in the discussion of Choudhury et al., 2020. This previous paper also shows that Kv3.3 is present in the postsynaptic MNTB neuron and contains several control experiments pertinent to the current manuscript which we have cited. We have added further data on Kv3.1 and Kv3.3 in the spiral ganglion and cochlear nucleus (figure supplements 2 and 3 to Figure 1, respectively). We also added to the discussion and we have conducted new experiments using Expansion Microscopy to show presynaptic and postsynaptic localisation of Kv3.3 and Kv3.1: see results and data in new Figure 2.

5. Figure 2: Could the increase in EPSC sizes be due to postsynaptic effects such as increased mEPSCs? This may be done by looking at asynchronous or spontaneous mEPSCs. In Figure 2, mini EPSCs should also be analyzed to help dissect the role of Kv3.3 at different stages of synaptic transmission.

We have added this data as a Supplementary Figure, Figure 3—figure supplement 1. There is no increase in mEPSC amplitude, if anything the quantal amplitude is decreased slightly in the Kv3.3KO – the opposite of what would be required to explain the increased evoked response. There is no evidence for other presynaptic changes, suggesting the increased transmitter release in the Kv3.3KO is purely an action potential based phenomenon. This is presented in the results and the discussion.

6. The potential effect of anaesthesia on in vivo recording in response to sound should be discussed. The FSL difference in the Kv3.3 KO may arise from spiral ganglia and CN neurons firing more slowly. These neurons also may express Kv3.1 and Kv3.3. Please emphasize this in Discussion.

Yes, part of the first spike latency (FSL) difference between WT and Kv3.3KO likely arises upstream of the MNTB, in the VCN and/or SGN, since both locations also express Kv3.1 and Kv3.3 (see supplemental Figures to figure 1, figure supplements 1-3). For a subset of our data we have assessed the FSL for the presynaptic calyceal action potentials, which relate to the output of VCN bushy cells. These data show that already at the level of the calyx, the Kv3.3KO latencies are significantly longer (WT Presynaptic FSL: 3.09 ±0.71ms, n=19 vs. Kv3.3KO presynaptic FSL: 6.05 ±2.79 ms, n=9, Mann-Whitney Rank Sum Test: p=0.002), which suggests that there is a cumulative effect of lacking Kv3.3 on FSLs along each synaptic relay of the auditory pathway, in which both the spiral ganglion neurons and the ventral cochlear are shown in the supplemental Figures to figure 1, figure supplements 1-3, as possessing Kv3.1 and Kv3.3.

Anaesthesia: To test specifically for the effect of Kv3.3 on the synaptic transmission between the calyx of Held to MNTB neuron, we compared the synaptic delay between the presynaptic calyceal action potential and the postsynaptic MNTB neuron AP between both genotypes. In WT, the synaptic delay was 0.45 ±0.05ms (n=26). This value fits perfectly into the range of synaptic delays recorded by different laboratories under different anaesthetics and in different animal models: ketamine/xylazine in mouse: 0.40ms (Blosa et al. 2015), 0.46ms (Kopp-Scheinpflug et al. 2003), 0.50ms (Lorteije et al. 2009); ketamine/xylazine in gerbil: 0.44ms (Tolnai et al. 2009). This narrow distribution of synaptic delays with different anaesthesia suggests that the effect of anaesthesia on synaptic delay is negligible compared to the effect of deleting Kv3.3, which resulted in a synaptic delay of 0.58 ±0.17ms (this manuscript). Taken together, there is always an effect of anaesthesia, just like there is always an effect of stress or attention if no anaesthesia is used. However, since we compare WT to KO within the same anaesthetic protocol; and since our WT data fit well to other published data, the genotype specific effect of the Kv3.3 on synaptic delay seems robust. We have added a paragraph discussing anaesthesia and the above comparisons to the discussion.

We have added two paragraphs to the discussion in which we discuss the evidence and logic for other Kv3 subunits contributing to presynaptic Kv3 channels. This also includes discussion of ionic versus non-ionic mechanisms for Kv3 roles in the terminal, and comments on areas that require further investigation.

Reviewer #1 (Recommendations for the authors):1. It follows that the decay of the presynaptic AP slope in Kv3.3KO correlated with a broadening of the AP waveform. It is curious that the AP waveform did not broaden with a Kv3.1 KO, yet the decay slope slowed almost to the extent of the Kv3.3KO. Could it be that Kv3.1 is acting at slightly different time-points during repolarization? What about the AP width at 75% or 25% of the amplitude?

See No.1 Essential revisions.

2. Figure 1B-C: the original study in JPhysiol (Forsythe, 1994) on the calyx AP shows that current step injections produce multiple APs in the presynaptic calyx, but just one in the postsynaptic MNTB cell. I thus wonder if maybe the recordings shown in Figure 1B-C of a solo AP may be perhaps from postsynaptic MNTB principal cells? Can the authors confirm with fluorescent dyes that the recordings shown in Figure 1B-C are indeed from the presynaptic calyx (e.g. they should show no mEPSCs)? Important to mention also that mice start to hear at about P12, so P10-12 calyces are still immature.

See No.2 Essential revisions.

3. Figure 6A: Kv3.1 KO show major effects in postsynaptic principal cell excitability and firing. This should perhaps be mentioned in the Abstract. Postsynaptic firing requires both Kv3.1 and Kv3.3.

We have already published a paper in The Journal of Physiology that shows that Kv3.1 and Kv3.3 both contribute to presynaptic APs repolarization in the MNTB. See Choudhury et al., 2020.

Reviewer #3 (Recommendations for the authors):I have a few points, which may be addressed by additional experiments and rewriting.1) The increase EPSC sizes are not due to postsynaptic effects such as increased mEPSCs? This may be done by looking at asynchronous or spontaneous mEPSCs.

This is covered by the essential revisions 5; new data has been added (see Figure3—figure supplement 1).

2) Kv3.3 channels are expressed in the postsynaptic cell, which may affect postsynaptic firing. One way will be to perform dynamic clamp in WT cells and apply synaptic conductance of WT and KO. Nevertheless, the authors may discuss this point carefully. Although it is likely that most effects are presynaptic, it is difficult for me to assess presynaptic affects precisely.

We agree that dynamic clamp studies would be helpful for future investigations of the role of Kv3 subunits in specific locations; location of the conductance is also a key issue and potential confound, so we have extended our knowledge of this issue with additional IHC data (Figure 2). We have focussed on the physiology in this manuscript, but we have added to the discussion in the interpretation of the presynaptic data and made clear that in any physiology, there will be interaction between the presynaptic and postsynaptic effects – indeed that was the purpose of the detailed investigation in Figure 7.

3) It is a bit difficult to understand why firing is reduced at the onset of sound stimulation in Kv3.3 KO mice. Overall, Fig7 is complex observation, and one needs some discussion.

This is now figure 8. The onset firing figure is the sum PSTH for all the data sets, as this best exhibits the increased variance in AP timing observed in the Kv3.3KO. Yes, it is hard to avoid this complexity and we appreciate the point; we have added to the Results section on page 9 and 10 and also included further discussion to help clarify the permutations of these observations.

4) Slice experiments and in vivo experiments use different ages. Is it reasonable to assume that the impact of Kv3.3 is the same?

We have added discussion and explanations of this to Essential revisions 3.